

# Combined North Atlantic and anthropogenic forcing of changes in the marine environments in the Gulf of Taranto (Italy) during the last millennium

Valerie Menke[1], Werner Ehrmann[2], Yvonne Milker[1], Swaantje Brzelinski[1,3], Jürgen Möbius[1], Uwe Mikolajewicz[4], Bernd Zolitschka[5], Karin Zonneveld[6], Kay Christian Emeis[1], Gerhard Schmiedl[1]

[1]Center for Earth System Research and Sustainability, Institute of Geology, University of Hamburg, Bundesstrasse 55, Hamburg, 20146, Germany
[2]Institute for Geophysics and Geology, University of Leipzig, Talstrasse 35, Leipzig, 04103, Germany
[3]Institute of Earth Sciences, Heidelberg University, Im Neuenheimer Feld 234-236, Heidelberg, 69120, Germany
[4]Max Planck Institute for Meteorology, Bundesstrasse 53, Hamburg, 20146, Germany
[5]Institute of Geography, University of Bremen, Celsiusstrasse 2, Bremen, 28359, Germany
[6]MARUM, University of Bremen, Leobener Straße, Bremen, 28334, Germany

*Correspondence to*: Valerie Menke (valerie.menke@uni-hamburg.de)

**Abstract.**

This study examines the multi-decadal to centennial variability of benthic ecosystems, depositional environments and biogeochemical processes in the Gulf of Taranto (Italy) over the last millennium. Our study is based on sediment cores from two sites in the eastern Gulf of Taranto (Mediterranean Sea), and benthic foraminifera data of 43 surface sediment samples from the western Adriatic Sea reflecting modern conditions. We use the data to unravel relative contributions of natural and anthropogenic forcing to conditions at the sediment-water interface in a marine setting with a long history of human impacts in river catchments. High abundances of infaunal foraminifera in surface sediments trace the nutrient-rich Po river outflow and display an area of high organic matter deposition in the north-eastern Gulf of Taranto. Decreasing Ca/Ti ratios suggest increasing terrigenous fluxes at ~1300 AD driven by wetter conditions during persistent negative phases of the North Atlantic Oscillation (NAO). A strong NAO connection is also evident in high-resolution clay mineral data. The smectite/illite ratio reflects variable Po river runoff, and correlates well with NAO strength for the past 300 years. Benthic ecosystem variability as reflected by foraminifera is closely linked to the Northern Hemisphere temperature evolution during the past millennium. Spectral analysis reveals a quasi-periodic variability of ~50 to ~70 years suggesting an Atlantic Multidecadal Oscillation (AMO) forcing of Italian hydrology. Coeval with increasing anthropogenic activity, the effects of rising temperatures and nutrient discharge during the past 200 years further enhanced nutrient and organic matter fluxes. This is reflected by a substantial rise in the abundance of shallow to intermediate infaunal benthic foraminifera (SIIBF) and a concurrent decrease of *Uvigerina mediterranea* $\delta^{13}$C since at least 1800 AD. The SIIBF decrease in the youngest samples likely reflects environmental effects of stricter regulations on fertilizer use in Italy and the reduction of sediment transport due to the stabilization of river banks.



## 1 Introduction

In recent years substantial progress has been made to understand large-scale climate trends of the Holocene and their impacts on marine and terrestrial environments based on the integration of proxy data and climate modelling (e.g., Chevalier et al., 2017; Grimm et al., 2015; Donders et al., 2008; Tierney et al., 2011). However, on shorter (i.e. decadal to multi-decadal) time scales, the understanding of regional patterns and their link to climate is still inadequate, partly because high-resolution proxy records are still sparse. The Mediterranean Sea is particularly suitable for investigating short-term and regional responses to climate changes, because it is a semi-enclosed basin with strong influence from land, located at the transition between the temperate mid- to high-latitudes and the subtropical low- pressure belt, and thus responds sensitively to climate variability (Lionello et al., 2006). But the Mediterranean borderlands have also been densely populated and subject to intense land use and agricultural activity during the past millennium (e.g., Holmgren et al., 2016; Lamb, 2013; Luterbacher et al., 2012; Pongratz et al., 2008). Therefore, the Mediterranean region is a highly complex system, and distinguishing between natural and anthropogenic influences on the oceanography and marine ecosystems is complicated but paramount for assessing future impacts of climate change.

The Mediterranean Sea responds to precession-driven changes in African Monsoon strength on orbital time scales (Cramp and O'Sullivan, 1999; Emeis et al., 2000; Rohling et al., 2015 and references therein; Weldeab et al., 2014). On suborbital timescales, Mediterranean climate dynamics are modulated by the North Atlantic climate variability, e.g. during the cooling of the Younger Dryas or the 8.2 ka event (Bar-Matthews et al., 1999; Cacho et al., 2001; Rohling and Pälike, 2005, and references therein). On shorter time scales, Mediterranean climate variability has also been influenced by fluctuations in the North Atlantic Oscillation (NAO) and the Atlantic Multidecadal Oscillation (AMO) in subtle, but nonetheless persistent, ways. While the NAO drives European winter temperatures and precipitation patterns (Beniston et al., 1994; Frisia et al., 2003; Hurrel, 1995, 2001), AMO fluctuations correlate with changes in summer temperature and precipitation (Enfield et al., 2001; Knight et al., 2006; Marullo et al., 2011; Sutton and Hodson, 2005).

Underlying these orbital and suborbital trends are changes in the activity of the sun reconstructed on the basis of atmospheric $\Delta^{14}C$ records (Stuiver, 1994). Solar forcing of Mediterranean climate is documented in various proxy records, e.g., central European lake level and Alpine glacier fluctuations (Holzhauser et al., 2005; Magny et al., 2007). A prominent example for effects of solar forcing is the rapid and short-term cooling of the Little Ice Age (LIA) that terminated the relatively warm and dry conditions of the Medieval Warm Period (MWP) (Despart et al., 2003; Grove, 2001; Guiot and Corona, 2010; Holzhauser et al., 2005; Luterbacher et al., 2001, 2004).

Combined evidence thus suggests that oceanographic and biogeochemical processes of the Mediterranean Sea are intimately linked to global and regional climate dynamics, but the relative influence of anthropogenic processes on late-Holocene marine environments is still a subject of debate. The Gulf of Taranto provides an ideal setting for addressing this question since it is especially susceptible to both climate variability and human-induced changes in land use, eutrophication and pollution on the Italian mainland. High amounts of nutrients and suspended sediments from the Po and other Italian rivers are transported NW-SE along the Adriatic coast and result in unusually high sedimentation rates on the outer shelf of the Gulf of Taranto. Sediment archives from these depocenters provide the basis for climate and environmental reconstructions on decadal to multi-decadal time scales (Grauel et al., 2013a; Taricco et al., 2015; Versteegh et al., 2007; Zonneveld et al., 2012). For example, Versteegh et al. (2007) suggested a centennial solar forcing in alkenone-based sea surface temperature



(SST) reconstructions from the Gulf of Taranto, implicating wind-induced mixing to play a crucial role for regional productivity. Goudeau et al. (2013) correlated high rates of detrital matter deposition on the eastern Apulian shelf with negative phases of the NAO and enhanced river discharge. Anthropogenic influences on the trophic state of the Po river discharge plume over the past 200 years is evident in dinoflagellate cyst distributions that responded to industrialization and fertilizer application in Italy (Zonneveld et al., 2012).

Benthic foraminifera are especially suitable for studying climatic and oceanographic processes since their abundance and faunal composition primarily depend on organic matter (OM) fluxes and oxygen availability at the sediment water interface; both factors are linked to atmospheric circulation and anthropogenic activity. Depending on the trophic state at the sea floor, they inhabit stratified microhabitats on and below the sediment surface and are commonly classified as epifaunal (living on top of, or up to 0.5 cm within the sediment), shallow and intermediate infaunal (commonly living between 0.5 and 2 cm in the sediment), and deep infaunal (commonly living below 2 cm in the sediment) (e.g., Corliss, 1985; Linke and Lutze, 1993; Lutze and Thiel, 1989; Mackensen and Douglas, 1989; Murray 2006; Jorissen et al., 1995). Changes in trophic conditions and the oxygenation of bottom waters can therefore be traced in the abundance of certain opportunistic species and shifts in the proportion of epi- and infaunal taxa (e.g. De Rijk et al., 1999; Jorissen et al., 1992, 1995; Schmiedl et al., 2000). Furthermore, the stable carbon isotope signature of benthic foraminifera is an important tool for assessing deep-water ventilation and for quantifying organic matter fluxes (e.g. Duplessy et al., 1988; Imbrie et al., 1984; Schmiedl et al., 2004; Shackleton 1987; Theodor et al., 2016 a,b; Zahn et al., 1986). Clay minerals and relative element abundances in sediments represent proxies for weathering on land, river discharge, and dispersal in the ocean that are independent of sea-floor ecology, allowing for a comprehensive reconstruction of past sedimentation processes and changes in climate and ocean circulation. The proportion of individual clay minerals in marine sediments mainly depends on the climatic conditions in the hinterland and the source rocks, making them ideal proxies for identifying source areas and dispersal pathways (e.g. Biscaye, 1965; Ehrmann et al., 2007; Petschick et al., 1996; Venkatarathnam and Ryan, 1971).

This study combines high-resolution benthic foraminiferal faunal and clay mineral data from two sediment cores in the Gulf of Taranto. Total organic carbon and nitrogen contents of bulk sediment and X-ray fluorescence elemental ratios provide additional information on organic matter fluxes and sediment origin and dispersal processes. Our multiproxy approach aims at identifying the relative contributions of natural climatic and anthropogenic forcing on benthic ecosystem variability and biogeochemical processes in the Gulf of Taranto over the past 1300 years.

## 2 Environmental Setting

The Gulf of Taranto is located at the southern tip of Italy (Fig. 1). The oceanography of the Gulf of Taranto is strongly influenced by water masses from the Adriatic Sea to the north and the Ionian Sea to the south. The Po river drains large parts of the southern Alps and northern Apennines and discharges high amounts of sediment- and nutrient-rich freshwater into the northern Adriatic Sea, from where it is transported to the south by the counter clockwise West Adriatic Current (WAC). The WAC (<37 psu; Lipizer et al., 2014) moves as a narrow coastal band along the western Adriatic margin and is further enriched in nutrients and sediment by smaller river systems of the Apennines before it reaches the Gulf of Taranto. The sediment supply from the southern Apennines is relatively small, but detectable near the coast through elevated smectite concentrations (Degrobbis



et al., 1986; Tomadin 2000; Milligan and Cattaneo, 2007) (Fig. 1 A). The stronger Padane detrital matter flux dominated by illite (from rivers Po, Brenta, Adige, Reno) is seen in a parallel band further in the basin (e.g. Milligan and Cattaneo 2007; Tomadin 1979, 2000) (Fig. 1 A). Maximum extension of the WAC plume along the coast and into the Gulf of Taranto occurs in late spring due to snow melt in the Alps and the northern Apennines,

and in fall when enhanced precipitation in the catchment areas leads to increased runoff. In the Gulf of Taranto, the WAC mixes with the warmer, oligotrophic and more saline Ionian Surface Water (ISW) (38.8-38.9 psu, 17–19 °C; Budillon et al., 2010) flowing into the gulf from the central Ionian Sea (Poulain, 2001; Turchetto et al., 2007) (Fig. 1 B). Annual mean SST in the Gulf of Taranto are ~18.7°C. The cold (13.5 °C; Artegiani et al., 1997), highly saline (>38.7 psu; Turchetto et al., 2007) and relatively nutrient-rich Levantine Intermediate Water

(LIW) flows into the Gulf of Taranto from the Levantine basin at a depth of 200–600 m (Fig. 1 B). Below the LIW, Adriatic Deep Water (ADW) consists of Northern Adriatic Dense Water (NadDW) formed by downwelling as a result of wintertime NE Bora wind outbreaks, and of Southern Adriatic Dense Water (SAdDW) formed by deep-water convection in late winter to early spring (Grbec et al., 2007; Turchetto et al., 2007).

Nutrient transport into the Adriatic Sea and the Gulf of Taranto is influenced by climate and precipitation with marked seasonal variations in river discharge. In summer, when the Inter Tropical Convergence Zone (ITCZ) migrates north, the Mediterranean is under the influence of the Hadley cell that creates hot and dry conditions (Alpert et al., 2006). In winter, the Hadley cell shifts south and the westerlies become the dominant driver of temperature and precipitation in the Mediterranean Sea region. Changes in the position of the moisture-rich

westerlies during winter are closely linked to the NAO. The NAO index reflects an atmospheric pressure gradient between the subtropical high-pressure cell over the Azores and the low-pressure cell over Iceland (Hurrel et al., 2001). A positive mode of the NAO index is connected to warm and wet winters over northern Europe, whereas a negative mode is connected to wet and relatively warm winters in southern Europe including the Mediterranean region (Ferrarese et al., 2008). The Atlantic Multidecadal Oscillation (AMO), expressed by

SST anomalies over the North Atlantic, is a mode of western European summer temperature and precipitation patterns with a periodicity of about 70 years (Enfield et al., 2001; Knight et al., 2006; Sutton and Hodson, 2005). The origin of the AMO is either linked to internal variability of the oceanic thermohaline circulation only, or to a free oscillation of the coupled ocean-atmosphere system (e.g. Delworth and Mann, 2000; Jungclaus et al., 2005). Cold and dry Bora winds blow in strong pulses over the Adriatic Sea in winter (Orlic et al., 1994; Rachev and

Purini, 2001) and enhanced wind-induced mixing of the oligotrophic ISW and the nutrient-rich LIW results in locally enhanced phytoplankton productivity. Northward winds called Siroccos bring warm and humid air from the Sahara to the Adriatic region year-round, but most commonly in spring (Sivall, 1957). Although Bora and Sirocco winds are short in duration, they can have substantial impact on sea level and circulation of the Adriatic Sea (Ferrarese et al., 2008; Jeromel et al., 2009; Orlic et al., 1994). An aeolian sediment flux related to Bora or

Sirocco winds is quantitatively negligible compared to the extensive sediment supply from local rivers (Chester et al., 1997).

**3 Methods**

Sediments from two sites on the eastern Apulian shelf were sampled during RV Poseidon cruise 411 in April 2011. Gravity core GeoB15403-4 and multicore GeoB15403-6 were recovered from site 03 (39°45.42 'N,



17°53.53' E; 170 m water depth); multicore GeoB15406-4 was recovered from site 06 (39°49.49' N, 17°50.02 'E; 214 m water depth) (Fig. 1 B). The 524.5 cm long gravity core from site 03 was sampled at 2.5 mm resolution for this study focusing on the uppermost 34 cm. GeoB15403-6 was sampled at 5 mm resolution over the length of 19 cm. The 34 cm long multicore from site 06 was sampled at 5 mm resolution. At both sites, sediments

consisted of homogenous, nannofossil-rich greyish to olive green muds. The core tops contain basically undisturbed surface sediments as indicated by the presence of an oxidized sediment layer. The oxygen penetration depth can be identified by distinct colour changes at ca. 6 cm for GeoB15403-4, and at 8 cm for GeoB15403-6. This suggests negligible sediment loss during core recovery. Oxygen penetration was down to 4 cm in GeoB15406-4. The difference between the stations is in good agreement with the eutrophic contrast

between both sites. We further analysed the distribution patterns of Recent benthic foraminifera in the upper 1-2 cm of multicores from 43 stations along the western coastline of the Adriatic Sea and the Gulf of Taranto to identify gradients in modern environmental conditions (Table S1). All surface samples were stained with rose Bengal after core retrieval to distinguish living from dead benthic foraminifera.

For faunal investigations, freeze-dried samples of GeoB15403-4 and GeoB15406-4, as well as the stained

surface samples were weighed and washed over a 63 μm sieve. The sub-fractions were dried at 40 °C. The coarse fraction was weighed and dry sieved over a 125 μm sieve before being split into subsamples containing on average 300 individuals. The identification of benthic foraminifera, preferentially on species level, was mainly based on the studies of Cimerman and Langer (1991), Jones and Brady (1994), Milker and Schmiedl (2012), Rasmussen and Thomson (2005) and Sgarella and Moncharmont Zei (1993).

Stable isotope ($\delta^{13}$C, $\delta^{18}$O) analyses of *Uvigerina mediterranea* (1-3 tests) *Cibicidoides pachyderma* (1-5 tests) from GeoB15406-4 were carried out at the Institute of Geophysics and Geology at the University of Leipzig. Carbonate powders were reacted with 105% phosphoric acid at 70°C using a Kiel IV online carbonate preparation line connected to a MAT 253 mass spectrometer. All carbonate values are reported in per mil relative to the Vienna PDB standard. Reproducibility was ensured by replicate analysis of the NBS19 standard

and was better than ± 0.029 ‰ for carbon and better than ±0.063 ‰ for oxygen isotopes.

The clay mineral analyses followed standard procedures (Ehrmann et al., 2007) and used bulk sediment samples from GeoB15406-4 and GeoB15403-6. Each sample was oxidized and disaggregated in a 5 % $H_2O_2$ solution. Then, carbonate was removed using 10 % acetic acid. The fine fraction was separated from the sand fraction by dry sieving through a 63 μm sieve. Settling tubes were used to separate the clay fraction (<2 μm) from the silt

fraction. The clay fraction was then analysed for its clay mineral composition using X-ray diffraction (XRF). We mounted the samples as texturally oriented aggregates and solvated them with ethylene-glycol vapour. A Rigaku Miniflex system with CoKα radiation (30 kV, 15 mA) was used for all analyses. Using a step size of 0.02°2θ and a measuring time of 2 s/step, the samples were X-rayed in the range 3–40°2θ. To better resolve the (002) peak of kaolinite and the (004) peak of chlorite, the range 27.5–30.6°2θ was analysed with a step size of

0.001°2θ and a measuring time of 4 s/step. The respective clay minerals were identified through their basal reflections. A semi-quantitative evaluation of the clay mineral assemblages was conducted by the use of empirically estimated weighting factors on integrated peak areas of the individual clay mineral reflections (Biscaye, 1964, 1965; Brown and Bindley, 1980). The respective clay mineral proportions of smectite, illite, chlorite and kaolinite are given in percent of the total clay mineral assemblage.




Total carbon and nitrogen were quantified in samples from GeoB15403-4 by a Carlo Erba Nitrogen Analyser 1500 (Milan, Italy). Particulate organic carbon was determined in weighed powdered bulk samples after acid treatments that removes carbonate. Precision is 0.05 % for total and organic carbon and 0.005 % for nitrogen.

GeoB15403-4 was scanned for the detection of major and trace elements with an ITRAX XRF-core scanner, COX analytical systems (Croudace et al., 2006), of GEOPOLAR at the University of Bremen. Sections were scanned applying a Mo-tube with a step size of 2 mm and a count rate of 10 s/per step. Tube settings were kept constant using a voltage of 30 kV and a current of 20 mA.

The age model for GeoB15403-4 is based on 11 analyses of surface-dwelling planktonic foraminifera by the AMS $^{14}$C method (Table 1). Dating was performed at Beta Analytic Inc. (Miami, Florida, USA). Radiocarbon dates were converted to calendar years using the MARINE13 database (Reimer et al., 2013) with a Delta R value of 73±34. Delta R was calculated using the MarineChrono reservoir database through an interpolation of data points close to the core locations. The $^{14}$C based age model was complemented by tephroanalysis of the upper 45 cm of the core (Menke et al., under review). Each sample was analysed for its glass shards content in vol% to identify primary tephra layers. Glass shards from the 63–125 μm fraction were analysed for major and minor elements following the procedure of Kutterolf et al. (2011) using a JEOL JXA 8200 wavelength dispersive electron microscope at GEOMAR, Kiel. The age model for GeoB15406-4 was calculated based on one AMS $^{14}$C date and a Delta R value of 73 ± 34 (Table 1).

Blackman-Tukey spectral analysis was performed on the original and detrended foraminiferal epifaunal and infaunal datasets of site 03. Both time series have been smoothed by subtraction of a 21-point running average to remove trends reoccurring on greater timescales. After resampling with Δ$t$ = 9.03 yrs, spectral analysis was performed using the software AnalySeries 2.0.8 (Paillard et al., 1996).

## 4 Results

### 4.1 Age Model

Results of AMS $^{14}$C dating are listed in Table 1 and are shown in Figure 2. Accordingly, core GeoB15403-4 has a basal age of ca. 55,600 yrs BP. Sedimentation rates increase from the older to the younger part of the core from 5.3 cm/kyr to 28.3 cm/kyr. Within the upper 38 cm, the sedimentation rates decrease to 23.47 cm/kyr. Due to the similar oxygen penetration depths in GeoB15403-4 and GeoB15403-6, we applied the age model of the gravity core also to the multicore. GeoB15406-4 has a basal age of ca. 236 yrs BP with a sedimentation rate of 114 cm/kyr.

### 4.2 Benthic Foraminifera

At site 03 and site 06, a total of 247 different foraminifera species were identified. The diversity at site 03 is generally higher and ranges from 52 to 95 taxa per sample compared to site 06 with 27 to 85 taxa. The most abundant species at site 03 are *Cassidulina laevigata s.l.*, *Melonis barleeanus*, *Gavelinopsis praegeri*, *Bulimina aculeata*, *Cassidulina crassa*, *Textularia pseudogramen*, *Miliolinella subrotunda*, *Globocassidulina subglobosa*, *Brizalina spathulata* and *Bulimina marginata*. The most abundant species at site 06 are *Cassidulina laevigata s.l.*, *Melonis barleeanus*, *Uvigerina mediterranea*, *Bulimina marginata*, *Brizalina spathulata*, *Cassidulina crassa*, *Gavelinopsis praegeri*, *Gyroidina umbonata*, *Bulimina costata* and *Quinqueloculina viennensis*). In order to characterize changes in food and oxygen supply over time, benthic foraminifera were grouped according to





their environmental preferences. Species preferring epifaunal and very shallow infaunal microhabitats and are known to be well adapted to oligotrophic to mesotrophic conditions with relatively low food supply and high oxygen availability (e.g., *Miliolida* spp.) are referred to as Epifaunal Benthic Foraminifera (EBF) (see Table S2 for details of classification). Foraminifera preferring mesotrophic environments with elevated food supply and

tolerance to moderate oxygen levels are referred to as Shallow to Intermediate Infaunal Benthic Foraminifera (SIIBF) (i.e. *Bulimina* spp., *Bolivina* spp., *Uvigerina* spp; Table S2). Species tolerant to eutrophic conditions and very low oxygen availability in the sediment pore water are classified as "Deep Infaunal Benthic Foraminifera (DIBF)" due to their ability to live deeper below the sediment surface (i.e. *Chilostomella oolina* and *Globobulimina pseudospinescens*; Table S2).

SIIBF are dominant at both sites with higher values at site 06 (41.6-67.4 %) after a steady and steep rise up to the most recent samples (Fig. 3 A and B). SIIBF at site 03 vary between 40.4 % and 56.3 % peaking at 1830 AD. Thereafter, values decrease and fluctuate around 50 %. EBF at site 03 vary between 9.9 % and 21 % without a clear long-term trend, while EBF at site 06 (5.3–23.9 %) begin at a similar concentration as at site 03 (Fig. 3 A), but steadily decrease to 5.3 % in the second to youngest sample. DIBF show the low concentrations at both sites

and vary between 0 % and 3.4 % at site 03, and between 0 % and 2.9 % at site 06. Although site 03 shows a slightly higher maximum than site 06, the overall abundance of DIBF over the course of the core is slightly higher at site 06. The $\delta^{13}$C of *U. mediterranea* at site 06 ranges between -1.07 ‰ and 0.36 ‰ and steadily decreases at the same rate in step with the decrease in EBF (Fig. 3 B). *Cibicidoides pachyderma* $\delta^{13}$C varies from -0.3 to 1.05 ‰ and follows the trend of *U. mediterranea*, but because individuals of *C. pachyderma* were not

abundant in every sample, the temporal resolution is smaller (Fig. 3 B).

A closer look at the dominant species at site 03 reveals that much of the high abundance after 1600 AD and the sharp peak at 1840 AD are based on rising relative abundances of *Bulimina* species, as well as *U. mediterranea* and *U. peregrina* (Fig. 4). Other dominant species, such as *M. barleeanus,* remain nearly constant, except during a sharp peak at 1787 AD and rising relative abundance in the most recent samples. Some species, such as *C.*

*laevigata* s.l. even decrease in relative abundance (Fig. 4). The latter species ranges between 6.3 % and 22.3 % and shows a general decrease towards the youngest samples, after a maximum abundance between ~800 and 1170 AD (Fig. 4). Site 06, covering ~230 yrs BP, shows different patterns in species abundance, even though most dominant species are the same (Fig. 5). *Cassidulina laevigata* s.l. (4.5-18 %) shows an overall trend to higher values in the younger samples, while *M. barleeanus* fluctuates around a relatively constant background

value and increases sharply (to 14 %) in the youngest sample. Similar to their patterns at site 03, *Bulimina* species and in particular *U. peregrina* and *U. mediterranea* are responsible for much of the rise in IIBF abundance at site 06. *Uvigerina peregrina* rises steadily from values around zero before 1800 AD to around 10 % in 1991 AD, following a steady rise of *U. mediterranea* relative abundances that peaks in the youngest sample at ~16 %.

Differences in species abundance between site 03 and site 06 are in good agreement with faunal distribution patterns in the surface sediments of the Gulf of Taranto (Fig. 6). Site 06 is located close to a region with a high abundance of intermediate and deep infaunal species. The main SIIBF species in both sediment cores (*Bulimina spp.*, *U. mediterranea* and *U. peregrina*) exhibit elevated abundances in surface sediments around the station at which the DIBF percentage is highest in the core record. The modern SIIBF abundance is more variable but high

values also concentrate in this area. Site 03 is located just at the southern border of this "infaunal hot spot", well represented by lower overall abundances of SIIBF including *Bulimina* spp., *U. mediterranea* and *U. peregrina.*



The relative abundance of *C. laevigata* is higher around site 03 and matches the higher overall abundance of this species in the down-core sediment record of this site when compared to site 06. *Melonis barleeanus* shows slightly higher abundances around site 03 but a more patchy distribution pattern with an exceptional high value at the north-easternmost surface station of the easternmost transect. For most surface stations, DIBF abundance

is anti-correlated to EBF abundance, which shows the highest value at the deepest station in the southeast. Elevated abundances of SIIBF can be traced along the western Adriatic Sea into the Gulf of Taranto where SIIBF values are generally high and also DIBF are most abundant (Fig. 6 B).

### 4.3 Clay minerals

The kaolinite concentration at site 03 fluctuates between 16 % and 20 %, while chlorite fluctuates between 8 %

and 12 %. Kaol/Chl ratios vary from 1.6 to 2.4 with a general increase from bottom to top (Fig. 7A). A marked shift from lower values of approximately 1.8 to higher values of approximately 2.2 occurs at ~1600 AD. Kaolinite concentration at site 06 varies between 18 % and 24 %, while chlorite varies between 8 % and 10 %. Kaol/Chl ratios fluctuate between 2.0 and 2.8, are constant at 2.4 before 1775 AD, reach a short maximum at around 1800 AD and then generally decrease in younger times (Fig. 7 A). Kaol/Chl ratios decrease from older to

the younger samples with exception of a strong rise in the youngest sample and a pronounced peak at ~1800 AD (Fig. 7 B).

Smectite concentrations at site 03 vary between 32 % and 41 %, and illite concentrations fluctuate between 31 % and 42 %. Sm/Ill ratios range between 0.8 and 1.3. The lowest values occur before 1300 AD. Sm/Ill is relatively high between 1300 and 1700 AD. In the younger part of the core the values generally decrease and show a strong

variability.

Smectite at site 06 varies from 31 % to 38 %, and illite varies from 32 % to 40 %. Sm/Ill ratios fluctuate between 0.9 and 1.3 (Fig. 7 B) showing a similar but overall lower range of values than Sm/Ill in the same time interval of site 3 (Fig. 7 A).

### 4.4 Organic Carbon and Nitrogen

Organic carbon at site 03 ranges between 0.5 and 0.8 % with one outlier of 0.9 % at 142.5 mm, which was excluded from our interpretation (Fig. 8). Values decrease relatively steadily from ~0.6 % to ~0.5 % between 900 AD and 1600 AD. After 1600 AD the organic carbon content rises continuously and exhibits highest values in the youngest samples. C/N ratios at site 03 range between 6.85 and 10.08 with the outlier of 12.5 at 142.5 mm (Fig. 8). The trend in the C/N ratios is similar to that in organic carbon content, with generally high values after

1600 AD and a maximum value in the youngest sample.

### 4.5 XRF scanning for element composition

Ca/Ti ratios vary between 11.8 and 17.3 with a steady decline from values around 16 in the older part of the core to the lowest value at 1820 AD (Fig. 8). After that, Ca/Ti ratios increase again until 1880 AD and level of around 13.5 in the youngest samples.



## 5 Discussion

### 5.1 Hydrological forcing of terrigenous matter fluxes to the Gulf of Taranto

The clay mineral composition documents the source areas and dispersal pathways of suspended terrigenous material from the Po and the Apennine rivers through the Adriatic Sea into the Gulf of Taranto. The offshore Po

(Padane) flux is rich in illite (> 50 %); in contrast, the coastal Apennine flux is characterized by a generally high smectite concentration (> 30 %) with the southern Apennines providing kaolinite-rich suspensions (Tomadin, 1979, 2000). Low Sm/Ill ratios therefore document a strong influence of the Padane flux. Minima occur around 1750 AD, 1470 AD and 1250 AD (Fig. 9 D). They coincide with high abundance of SIIBF (Fig. 9 B), which implies that more nutrients and suspended matter at times of enhanced Po river outflow lead to more eutrophic

conditions in the Gulf of Taranto and foster the development of shallow and intermediate infaunal microhabitats. The drop of Po-derived suspension at 1300 AD roughly coincides with the onset of the LIA and glacier advances in the Alps (Holzhauser et al., 2005; Le Roy et al., 2015). The cool climate possibly led to increased accumulation of ice and snow in the Alps and to overall low Po river runoff and sediment discharge. While Sm/Ill ratios forced by the Po runoff appear to have been limited by the cool climate, the Kaol/Chl ratios rise

steadily since 1300 AD and follow the general trend towards a more negative NAO after 1300 AD (Fig. 9 H and I). This can be explained by the fact that the southern Italian borderlands and Apennine mountains are less susceptible to a colder climate due to their lower elevation and location further south (Drever and Zobrist, 1992; Matsuoka, 2008; Cyr et al., 2010). A negative NAO index corresponds to warmer and wetter winters across southern Europe and the Mediterranean Sea (Hurrel, 1995; Tomasino and Dalla Valle, 2000; Trouet et al., 2009),

which explains the observed increase in discharge and sediment transport from the southern Apennines and the borderlands of the Gulf of Taranto. The abundance of the shallow infaunal *U. mediterranea* slightly rises simultaneously arguing for a steady increase in organic matter fluxes since ~1300 AD and a relatively high OM quality (Fig. 4) (Fontanier et al., 2002; Koho et al., 2008). A close relation between NAO strength and precipitation in the Po river catchment has been confirmed based on direct measurements of the NAO index and

Po discharge since 1921 (Tomasino and Dalla Valle 2000). It is also evident by the correlation of high proportions of illite (low Sm/Ill ratios) with a negative NAO index, as illustrated in the high-resolution record for the past 300 yrs at site 06 (Fig. 10 A and B). Although at site 06 the Sm/Ill ratios trace fluctuations in the NAO index well within the accuracy of the age model, the reliability of different NAO reconstructions has to be questioned because of significant discrepancies between reconstructed and measured NAO indices of the past

~200 years (Fig. 10 A black and red lines, e.g. Jones and Mann, 2004). We therefore argue that a comparison of the NAO index and proxy data can provide useful insights into North Atlantic climate links for the past 200 years, while applications predating this period are less confident. The longer Sm/Ill record at site 03 is most likely overprinted by the strong temperature influence on large areas of the Po river catchment during the LIA. The steady decline in southern Apennine suspension load since ~1800 AD, evident through lower Kaol/Chl

ratios at site 06, is likely attributed to a growing dominance of Po-derived material since the end of the LIA (Fig. 10 E). This interpretation appears reasonable, because the Po catchment area is much larger when compared to that of the Apennine rivers. Accordingly, Po suspension loads increased proportionally to precipitation at times of a persistent negative NAO mode and generally increasing temperatures.

A strong increase in terrigenous sediment input around 1300 AD, simultaneous to rising Kaol/Chl ratios, is also

documented by the Ca/Ti ratios obtained from site 03 (Fig. 9 G). Previous studies in the Gulf of Taranto based



on XRF core scanning showed that changes in Ca/Ti primarily reflect the relative dominance of terrigenous minerogenic versus marine biogenic input (Goudaeu et al., 2014; Richter et al., 2006; Rothwell and Rack, 2006). These studies further demonstrated a positive correlation between low Ca/Ti values, high terrestrial matter input and a negative NAO index for the past 16 kyrs. They also documented the same steep rise in terrestrial matter

input during the past 1000 years that is also found in our data.

To summarize, the combination of clay mineral and XRF data concordantly suggests an increase in terrigenous fluxes to the Gulf of Taranto at ~1300 AD. This change in sedimentation can be attributed to generally wetter conditions and enhanced runoff from the borderlands of the Gulf of Taranto and is correlated to persistent negative NAO phases. The Po river runoff shows a strong correlation to NAO strength for the past 300 years.

However, runoff declined at the onset of the LIA during which only strong runoff events had an impact on the benthic foraminifera record.

**5.2 Northern hemisphere climate forcing on marine ecosystems of the Gulf of Taranto**

The pronounced mobilization of terrestrial material and its deposition in the Gulf of Taranto around 1300 AD is not reflected in the abundance and species composition of the benthic foraminiferal fauna (Fig. 9 B and G).

Obviously, the fauna did either not respond to the associated changes in substrate or these changes were negligible. Instead, the trend in the relative abundance of SIIBF taxa generally follows the Northern Hemisphere temperature curve reconstructed from tree ring, ice core, coral and sediment records of the past 1500 years (Mann et al., 2009) (Fig. 9 A). Numerous previous studies demonstrated that the abundance of shallow and intermediate infaunal foraminifera is closely related to the availability of food (e.g., De Rijk et al., 2000; Jorissen

et al., 1995; Schmiedl et al., 2000), which in the Gulf of Taranto is controlled by riverine nutrient sources and related surface water productivity (e.g. Focardi et al., 2009; Zonneveld et al., 2009;). This suggests that positive northern hemisphere temperature anomalies resulted in enhanced precipitation in the catchment areas of the Italian rivers in the borderlands of the Adriatic Sea and the Gulf of Taranto. The positive correlation between temperature anomalies and Po river discharge during the instrumental period since 1920 AD (Tomasino and

Dalla Valle, 2000) corroborates this interpretation. The C/N ratios in our cores increase simultaneously with the abundance of the SIIBF at around 1600 AD (Fig. 9 B and C). The two curves also show similarities during other time intervals, for example a rise around 1350 AD and minimum values between 1500 and 1600 AD, followed by a steep increase. Increased C/N ratios at 1600 AD suggest enhanced input of terrestrial organic matter (Meyers, 1994) and Kaol/Chl and Sm/Ill ratios show a concurrent steep rise of suspended terrigenous material.

These observations support our conclusion of elevated precipitation and associated runoff at times of enhanced Northern Hemisphere temperatures (Fig. 9 D and H).

Other proxy data from this region documented a similarly strong temperature component in their records. Frisia et al. (2003) argue, that their stalagmite record is not predominantly reflecting variations in precipitation but also temperature. Low winter temperatures limited stalagmite growth and pronounced lamina growth started at about

1850 AD, when Northern Hemisphere temperatures began to rise steeply after the LIA. Alkenone-derived sea surface temperatures (SST) from the Gulf of Taranto show a similar temperature trend (Fig. 9 F; Versteegh et al., 2007). The SST record corresponds to changes in atmospheric $\Delta^{14}C$ for the period between 1420 AD and 1920 AD, the time before human interferences have likely substantially overprinted the natural climate signal (Fig. 9 E and F). Some short-term fluctuations in the SST and benthic foraminiferal records, however, do not correspond

to changes in the Northern Hemisphere temperature record. The most prominent example is a SST and infauna



maximum between 1650 AD and 1850 AD. Since the change in solar forcing is too small to directly influence regional SST, Versteegh et al. (2007) imposed wind stress to be a key factor, because it influences sensible and latent heat exchange at the water-air interface and strongly modulates the surface-water productivity via turbulence. Turbulence causes deeper convection (Grbec and Morovic, 1997), which in turn cools the surface

water and introduces nutrients into the photic zone (Boldrin et al., 2002). This process could expand the season of alkenone production from late autumn to early spring when overall temperatures are higher and could also cause the peak in alkenone temperatures between 1650 AD and 1850 AD. Enhanced primary production and related organic matter fluxes to the sea-floor is a plausible explanation for the peaks in the SIIBF abundance in both sediment cores (Fig. 9 B and Fig. 10 C) indicating a temporal shift to more eutrophic conditions. The higher

temporal resolution at site 06 allows to identify smaller peaks within the overall rise in SIIBF abundance that coincide with high sunspot numbers supporting the hypothesis of sunspot variations influencing seafloor ecosystems through surface productivity (Fig. 10 D). However, at the time of SST, SIIBF and C/N maxima, the Sm/Ill ratios indicate elevated Po river input. This implies that the eutrophic conditions were caused by enhanced Po river outflow, possibly due to the combined effects of a negative NAO index and rising temperature.

Power spectra of the detrended time series of the different benthic foraminiferal microhabitat groups at site 03 reveal a significant periodic variability on multi-decadal time scales, centered at ~54 yrs and ~72 yrs (Fig. 11). According to Knudsen et al. (2011) a 55- to 70-year quasi-persistent AMO existed through most of the Holocene and was forced by internal ocean-atmosphere variability. The AMO coupling to regional climate appears to have been modulated by insolation-driven shifts in atmospheric circulation patterns and sea-ice cover. Sutton and

Hodson (2005) proposed that warm phases of the AMO led to elevated summer temperatures and precipitation over Western Europe during the 20th century. A study on multi-decadal variability of Mediterranean SST's (Marullo et al., 2011) suggests a 70 yr periodicity suggesting a stimulation by the AMO. Our benthic foraminiferal record reveals striking similarities with the AMO reconstruction (Mann et al., 2009) and supports the hypothesis that the AMO influenced the central Mediterranean hydrology and marine environments during

the past 1100 years.

Although the fluctuations of the different mineralogical (Ca/Ti, clay mineral ratios), biogeochemical (C/N ratio) and foraminiferal proxy records can be generally reconciled with regional precipitation changes and hemispheric climate modes, a detailed comparison reveals a decoupling of the terrigenous fluxes and faunal development for most species (Figs. 9 and 10). One explanation could be that changes in terrestrial input due to elevated

precipitation at times of a negative NAO were too subtle to influence substrate composition, organic matter availability and biogeochemical gradients in the sediment. This seems likely when comparing the relatively low-amplitude changes in clay mineral ratios at site 03 to the pronounced shifts observed in sediment cores from the Aegean Sea across major transitions such as the last glacial termination (Ehrmann et al., 2007). We argue that while the NAO signal seems to modulate the overall hydrological regime, the AMO paces rainfall and related

impacts on marine environments on multidecadal time scales.

### 5.3 Anthropogenic impact on marine environments of the Gulf of Taranto

At least since the time of the Roman Classical Period anthropogenic activity had a significant impact on the depositional dynamics of the Po River and on the western Adriatic Sea. At that time erosion and sediment input increased due to deforestation, agriculture and river embankment (Stefani and Vincenzi, 2005). The demise of

the Roman infrastructure after the fall of the Roman Empire resulted in the re-establishment of near-natural



conditions over large areas of Italy. This was around 450–650 AD, a time when climate was cooling (Luterbacher et al., 2012). Between 1465 AD and the end of the 1500s, large hydraulic construction projects were completed in order to protect the local agriculture. Negligence of these constructions combined with strong storm surge events and disastrous flooding, which characterized the LIA, resulted in a loss of reclaimed

territories and a transfer back to marshland between the 17[th] and the first half of the 19[th] century (Simeoni and Corbau, 2009, and references therein). Increasing population and anthropogenic activity in the Po Valley since at least the 1600s (Lahmeyer, 2006; Lotze et al., 2011; McEverdy and Jones, 1978) most likely resulted in enhanced discharge of suspended matter, nutrients and trace elements into the northern Adriatic Sea. This eutrophication caused intense and recurrent local plankton blooms and anoxia (e.g., Degobbis et al., 2000;

Giordani and Angiolini, 1983; Justic, 1987; Marchetti et al., 1989).

     The quality of the plume waters has also an impact on the composition of dinoflagellate cyst associations (Zonneveld et al., 2012), and the accumulation rate and abundance of different dinoflagellate species registering distinct historical events and inventions, such as the beginning of the industrial revolution in Italy, the introduction of ammonia as fertilizer, and the recent restriction of fertilizer use within the Po Valley (Figs. 9 and

10, orange triangles).

     Persistent dominance of shallow to intermediate infaunal benthic foraminifera in the Gulf of Taranto since ~1800 AD also expresses eutrophication by Po riverine nutrients, but the faunas at the two study sites differ in the proportions of shallow infaunal and epifaunal foraminifera. The relative abundance of SIIBF at site 03 declines after a peak at ~1800 AD, whereas the SIIBF at site 06 continues to rise until about 1970 AD (Figs. 3 and 9).

Also, the proportion of the epifauna at site 03 shows only minor variations, while it continuously declines at site 06. Sedimentation rate at site 06 is about three times higher than at site 03. Varying sedimentation rates between different stations in the Gulf of Taranto have also been reported by Goudeau et al. (2014) supporting earlier evidence for regionally heterogeneous sediment accumulation in this region (Rossi et al., 1983). The difference in the depositional environment between the two sites is today reflected in the distribution pattern of modern

benthic foraminifera in surface sediments of the Gulf of Taranto when compared to the Adriatic Sea (Fig. 6). Elevated abundances of SIIBF taxa and the development of deep infaunal niches (Jorissen et al., 1995) around site 06 suggest the presence of high organic matter availability at the sea floor. This eutrophic hotspot spreads across isobaths (Fig. 6; Zonneveld et al., 2008) and thus is not directly linked to water depths, but to more or less stationary small-scale eddies in the surface water causing zones of nutrient injection and enhanced primary

productivity in the surface ocean (Pinardi et al., 2016) and related organic matter fluxes to the sea-floor. Small-scale variability and local patchiness of high-productivity zones in the Gulf of Taranto are also indicated by satellite images of chlorophyll-a concentration in the upper ocean (Zonneveld et al., 2009). The associated regional slow-down of current velocities induces deposition of suspended matter and leads to enhanced accumulation of organic matter at the sea-floor. Accordingly, the fauna comprises both less opportunistic taxa,

such as *U. mediterranea,* and more opportunistic taxa, such as *U. peregrina* (Fontanier et al., 2002; Koho et al., 2008). The latter species benefits from seasonal pulses of freshly deposited phytodetritus and its abundance mimics the seasonal dynamics of surface water productivity as well as the total amount of riverine nutrient fluxes. This makes *U. peregrina* a suitable indicator species for the human-induced eutrophication in the Gulf of Taranto that started around 1800 AD or even earlier. The abundances of *Bulimina* species (Figs. 4 and 5)

resemble the trends seen in *U. peregrina* and argue for similar controls by organic matter fluxes.



Independent micropaleontological and biogeochemical data corroborate the inference of enhanced nutrient input into near-coastal Italian surface waters and associated higher organic matter fluxes in the Gulf of Taranto during the past two centuries. The decrease in the $\delta^{13}$C record of the shallow infaunal *U. mediterranea* corresponds to a similar trend in the $\delta^{13}$C record of the epifaunal *C. pachyderma*. Therefore, we assume that *U. mediterranea* is

suitable to display the general source of the bottom water mass for a first approximation, although it is a shallow infaunal species whose isotopic signature is commonly influenced by the pore water isotopic gradient (Linke and Lutze, 1993; Schmiedl et al., 2004) (Fig. 3 B). Accordingly, $\delta^{13}$C values of *U. mediterranea* can be used to trace the influence of the nutrient-rich WAC (Grauel et al. 2013b). The $\delta^{13}$C values of *U. mediterranea* decrease around 1770 AD towards the younger part at site 06 and suggest increased organic matter fluxes and associated

remineralisation rates (Theodor et al., 2016a, b) (Fig. 10). The inferred elevated nutrient input is consistent with increasing concentrations of plant waxes suggesting an enhanced supply of terrestrial plant material into the Gulf of Taranto starting around 1800 AD (Grauel et al., 2013 b). Concurrent with the pronounced maxima in the SIIBF at ~1850 AD in both of our sediment cores, epiphytic benthic foraminifera diminished around the Po delta as response to higher nutrient loads between 1840 and 1870 AD (Barmawidjaja et al., 1995).

The dinoflagellate abundance started to increase in the 1830s and specific species, which prefer eutrophic conditions, increased again around 1930 AD (Sangiorgi and Dongers, 2004). In a more recent study, Zonneveld et al. (2012) used the abundances of dinoflagellate cysts to document changes in the nutrient status of surface waters in the Golf of Taranto that responded to the industrial revolution and the introduction of ammonia as fertilizer in Italy in 1890 and 1920, respectively. In addition, the abundance of coccolithophores started to

increase from the late 1800s, followed by the increase of diatoms and other siliceous plankton in the 20[th] century (Puskaric et al., 1990). We conclude that combined evidence from our own and other biogeochemical and micropaleontological studies document a pronounced human-induced eutrophication of marine environments along the Italian coast of the Adriatic Sea and the Gulf of Taranto since approximately 1800 AD. Our new data demonstrate that the impacts of changes in Italian land-use are not restricted to near-coastal surface water

ecosystems, but can be traced even in outer shelf benthic ecosystems, a case of strong regional land-ocean linkages and tight bentho-pelagic coupling.

The decline of the shallow infauna in the youngest sediments of both cores (Figs. 9 and 10) reflects decreasing surface water productivity that may be the result of strict environmental regulations and the decline in fertilizer use in Italy since the 1970s (Zonneveld et al., 2012). But it is unclear whether the observed decrease in

chlorophyll-a concentrations over the last decades is due to environmental legislation, changing population dynamics or to natural causes such as changes in the precipitation patterns in the river catchments (Mozetic et al., 2010). Over the same time span there is evidence for a reduction of sediment transport due to reforestation and stabilization of lower order channels, and the construction of embankments and dams along the course of the Po River (Galay, 1983; Herget 2000; Marchetti et al., 2002 and references therein; Rodolfi, 1988). The input of

sediment particles from the Po river to coastal environments in the Adriatic has almost completely ceased since the mid-20[th] century because of dam construction, measures against soil erosion, and massive legal and illegal river bed excavation (Stefani and Vincenzi, 2005).





**6 Conclusions**

Benthic ecosystem variability in the Gulf of Taranto is driven by nutrients from the Po and to a lower extent from Apennine rivers that are transported along the Italian east coast with the WAC. A high-accumulation depocenter is developed at the distal end of the WAC in the northeastern Gulf of Taranto. Enhanced
sedimentation rates in this area are accompanied by eutrophic conditions reflected by regionally high abundances of shallow and deep infaunal benthic foraminifera at the sea floor.

Clay mineral and XRF data provide evidence for an increase in terrigenous fluxes to the Gulf of Taranto at ~1300 AD driven by generally wetter conditions and enhanced river runoff from the borderlands of the gulf at times of persistent negative NAO phases. Elevated Po River runoff correlates to NAO strength for the past 300
years. However, during the LIA overall river runoff declined and only strong runoff events had an impact on the trophic state of benthic ecosystems in the Gulf of Taranto.

Benthic ecosystem variability on the outer shelf of the Gulf of Taranto is closely linked to the Northern Hemisphere temperature record. Positive temperature anomalies most likely resulted in enhanced precipitation delivering nutrients and organic matter to the gulf. Spectral analysis of benthic microhabitat groups reveal a
periodicity of ~50 to 70 years suggesting an AMO forcing of benthic ecosystems through elevated summer temperatures and precipitation in the river catchment. Our results suggest, that the NAO determines the overall hydrological regime but the AMO paces rainfall and marine ecological and biogeochemical responses on multi-decadal timescales.

The effect of rising temperatures and nutrient transport during the past 200 years is amplified by increasing
anthropogenic activity since the 1600s. Enhanced nutrient input and higher organic matter fluxes transported by a strong WAC resulted in increasing SIIBF abundance (especially the opportunistic *U. peregrina*), and since ~1800 AD led to a decrease of $\delta^{13}$C values of the shallow infaunal *U. mediterranea*. The decline of the SIIBF abundance during the past few decades is likely the result of stricter regulations on fertilizer use in Italy and the reduction of riverine suspension load due to the stabilization of river banks.

**Acknowledgements**

We thank the master and crew of "RV Poseidon" for excellent collaboration. We thank S. Haeßner for support with sample preparation for clay mineral analyses, Niko Lahajnar and Frauke Langenberg for biogeochemical analyses and Stefan Krüger for performing the stable isotope measurements. This research was funded by the Deutsche Forschungsgemeinschaft, grant SCHM1180/18 and the "School of Integrated Climate System Science
(SICSS)" of the Cluster of Excellence "Integrated Climate System Analysis and Prediction (CliSAP)".

The raw data to this paper will be available at http://dx.doi.org/......../PANGAEA after acceptance of the paper for publication.

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



**Table 1: Age tie points used for generating the age models for sediment cores GeoB15403-4 (site 03) and GeoB15406-4 (site 06).**

| Site | Depth (mm) | Radiocarbon age (BP) | Calendar years (cal BP) | Data |
|------|-----------|----------------------|-------------------------|------|
| 03 | 0.00 | | -61 | Sediment Surface |
| | 211.25 | 1380±30 | 839±101 | AMS[14]C dating |
| | 358.00 | | 1174 | Lipari X tephra (a) |
| | 881.25 | 2320±30 | 1846.5±122.5 | AMS[14]C dating |
| | 1248.75 | 2710±30 | 2316±142 | AMS[14]C dating |
| | 1598.75 | 3410±30 | 3191±138 | AMS[14]C dating |
| | 1845 | 4020±30 | 3947±132 | AMS[14]C dating |
| | 1983.75 | 4270±30 | 4277±133 | AMS[14]C dating |
| | 2181.25 | 5470±30 | 5767±116 | AMS[14]C dating |
| | 2391.25 | 6290±30 | 6656.5±124.5 | AMS[14]C dating |
| | 2635.00 | 8350±40 | 8807.5±170.5 | AMS[14]C dating |
| | 2868.75 | 9920±30 | 10808±169 | AMS[14]C dating |
| | 4061.25 | 29550±110 | 33286.5±356.5 | AMS[14]C dating |
| 06 | 0.00 | | -61 | Sediment Surface |
| | 307.50 | 670±30 | 208±126 | AMS[14]C dating |

References: (a) Keller, 2002, Albert et al., 2012, Menke et al., under review



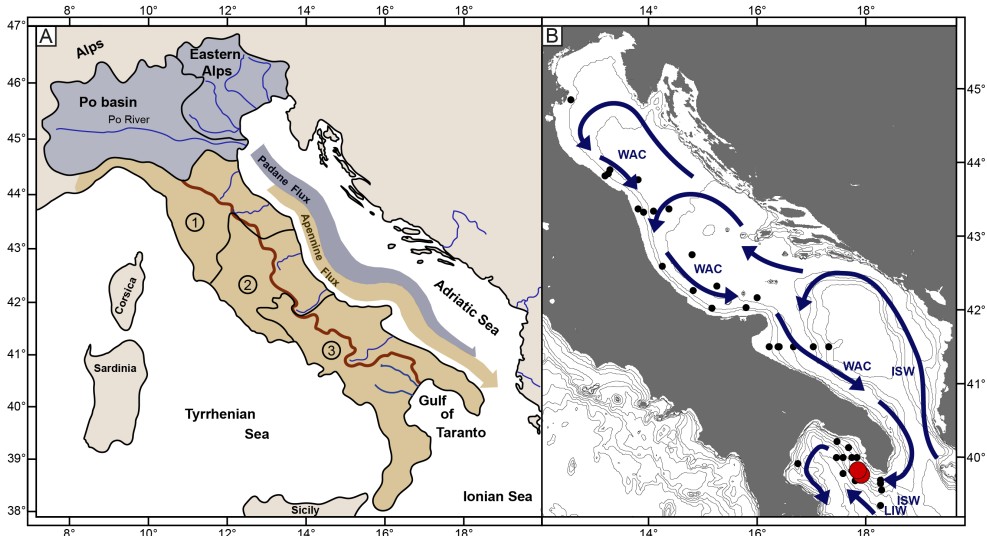

**Figure 1: A) Large scale overview of Italian catchment areas and dispersal pathways of the different freshwater sources. The coastal Apennine flux (brown) is fed by several smaller Apennine rivers and flows parallel to the Padane flux (grey) which carries material from the Po valley and the eastern Alps. Numbers denote Apennine areas including the northern (1), middle (2), and southern (3) Apennines. Red bold line displays the border of the Apennine river catchments. B) Location of the sediment cores investigated in this stiudy. Black dots display the position of surface sediment samples. The red dot on the lower right shows the locations of core sites GeoB15403-4 and -6 (site 03) and upper left red dot shows the location of GeoB15406-4 (site 06) in the Gulf of Taranto. Black arrows show the flow directions of the West Adriatic Current (WAC), The Adriatic Surface Water (ASW), the Ionian Surface Water (ISW) and the Levantine Intermediate Water (LIW).**



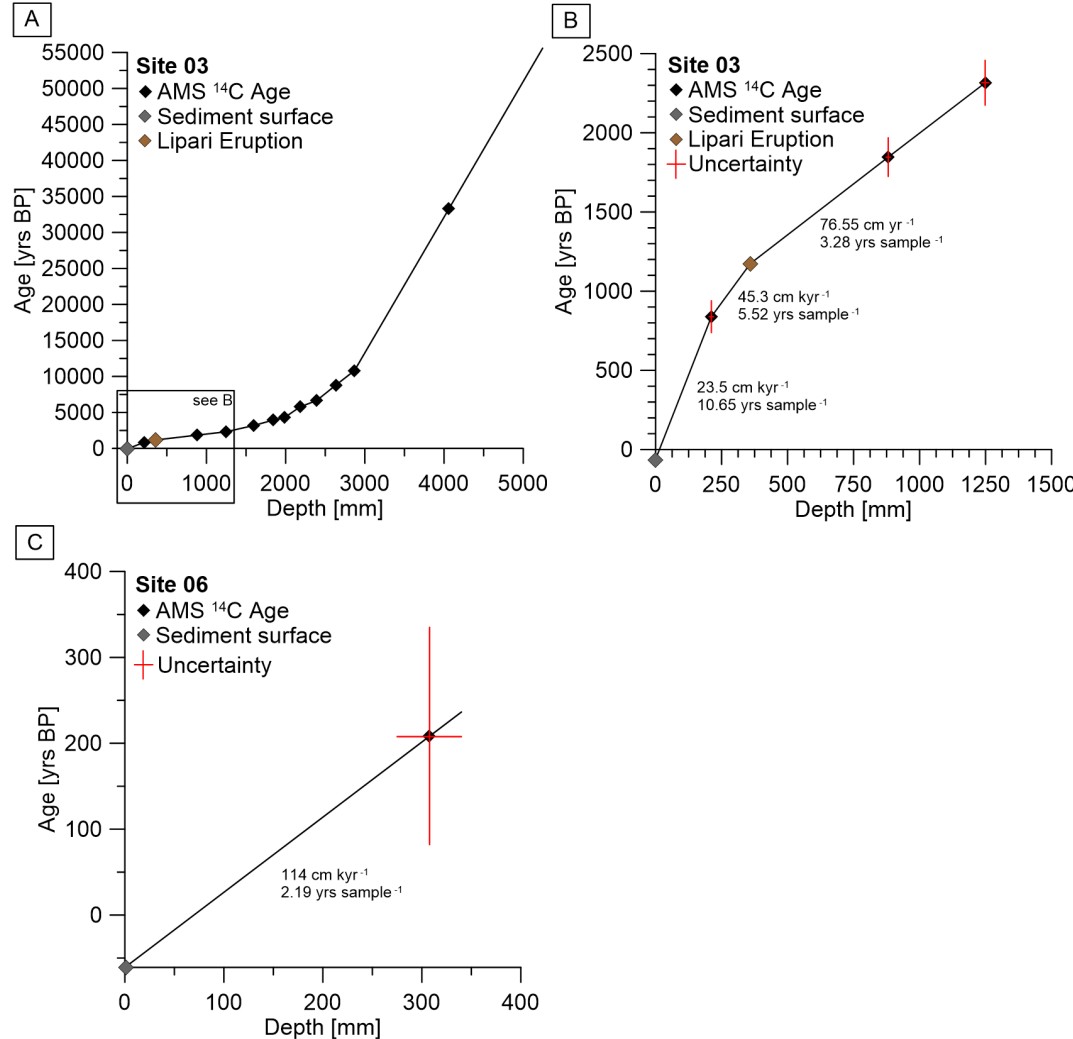

**Figure 2: Age versus depth model for gravity core GeoB15403-4 from site 03 and multicore GeoB15406-4 from site 06.
A)** Age model for the entire sediment succession of core GeoB15403-4 from site 03. **B)** Detail of age data for the upper
1.5 m of core GeoB1503-4 from site 03 including interpolated sedimentation rates and sample resolution. **C)**
Age/depth model for GeoB15406-4 from site 06 including numbers for interpolated sedimentation rate and sample
resolution. Vertical red bars in B and C indicate 2σ uncertainties of radiocarbon ages, horizontal red bars indicate the
depth interval from which planktonic foraminifera were selected for AMS $^{14}$C dating.

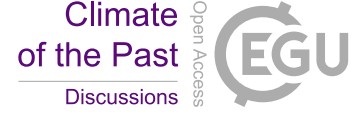

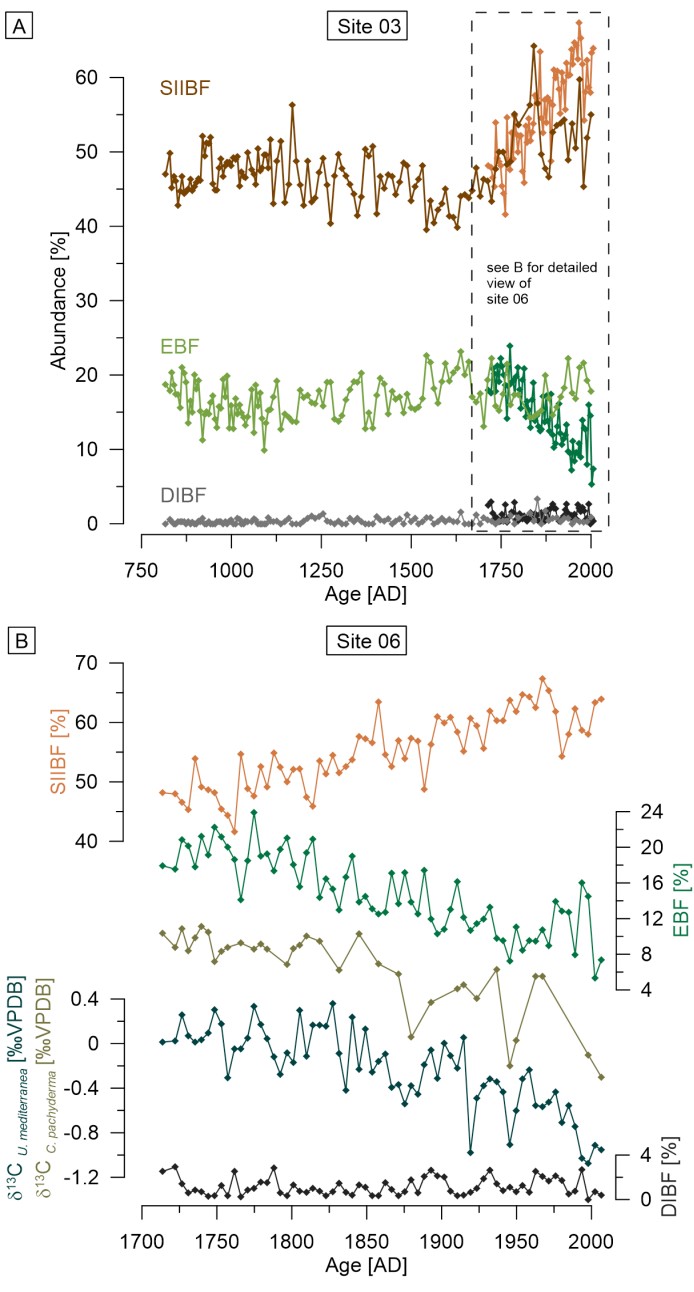

**Figure 3: A: Relative abundance of benthic foraminiferal microhabitat groups versus time including EBF: Epifaunal Benthic Foraminifera (bright green for site 03, dark green for site 06), SIIBF: shallow to intermediate infaunal benthic foraminifera (brown for site 03, orange for site 06), and DIBF: Deep Infaunal Benthic Foraminifera (grey for site 03, black for site 06). B: Detailed view of microhabitat groups at site 06 (colours as in A) and δ¹³C of *C. pachyderma* (olive green) and *Uvigerina mediterranea* (dark green) versus time.**



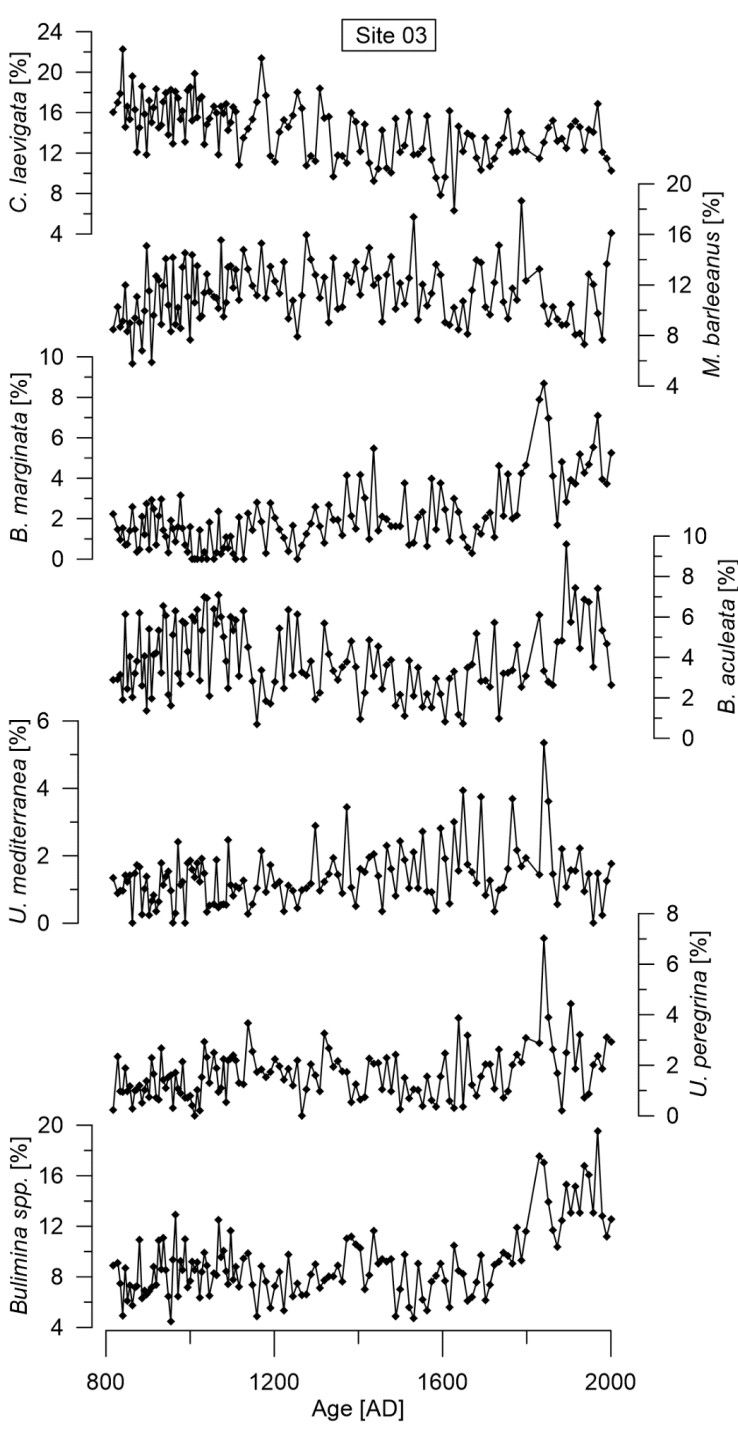

**Figure 4: Relative abundance of dominant benthic foraminifera taxa versus time at site 03.**





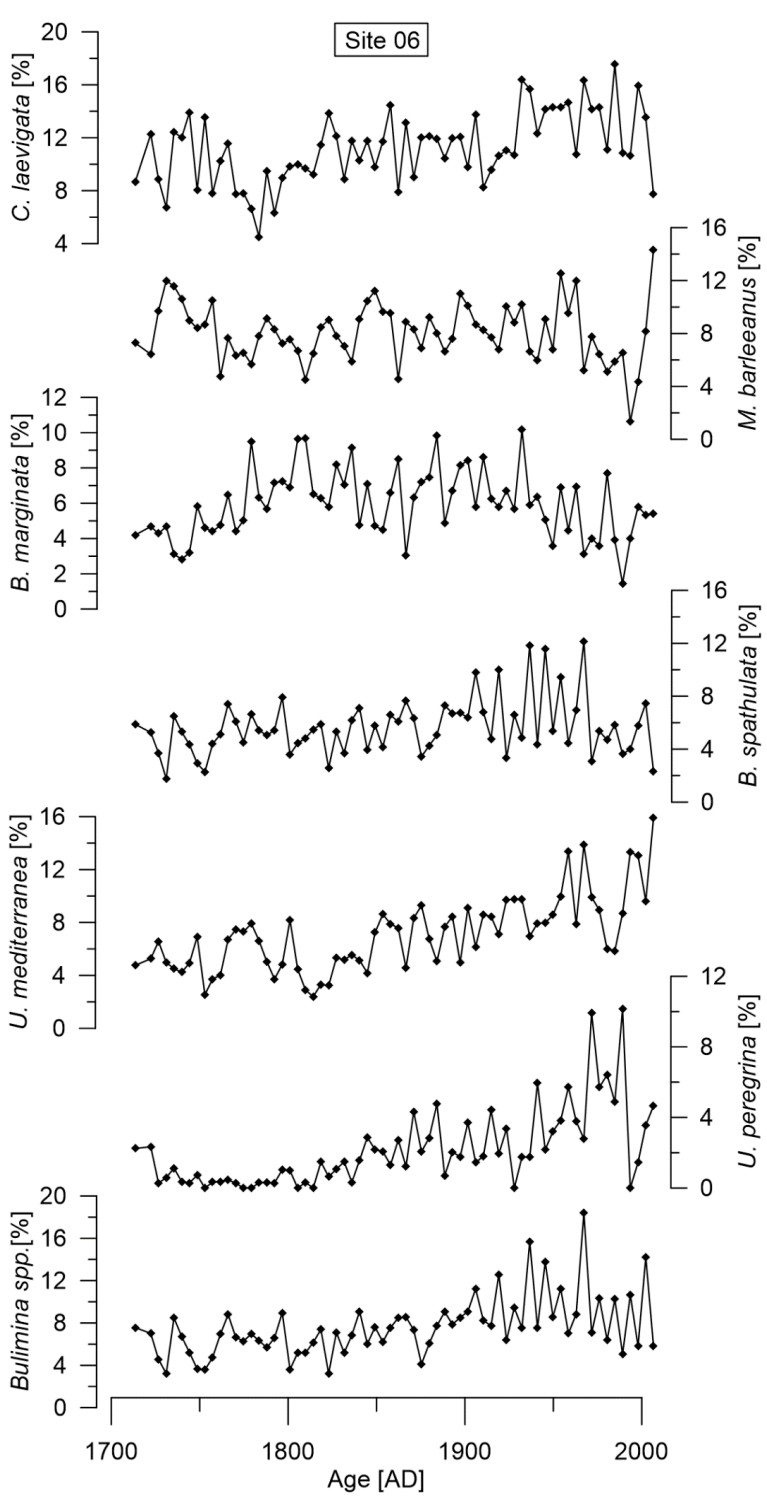

**Figure 5: Relative abundance of dominant benthic foraminifera taxa versus time at site 06.**



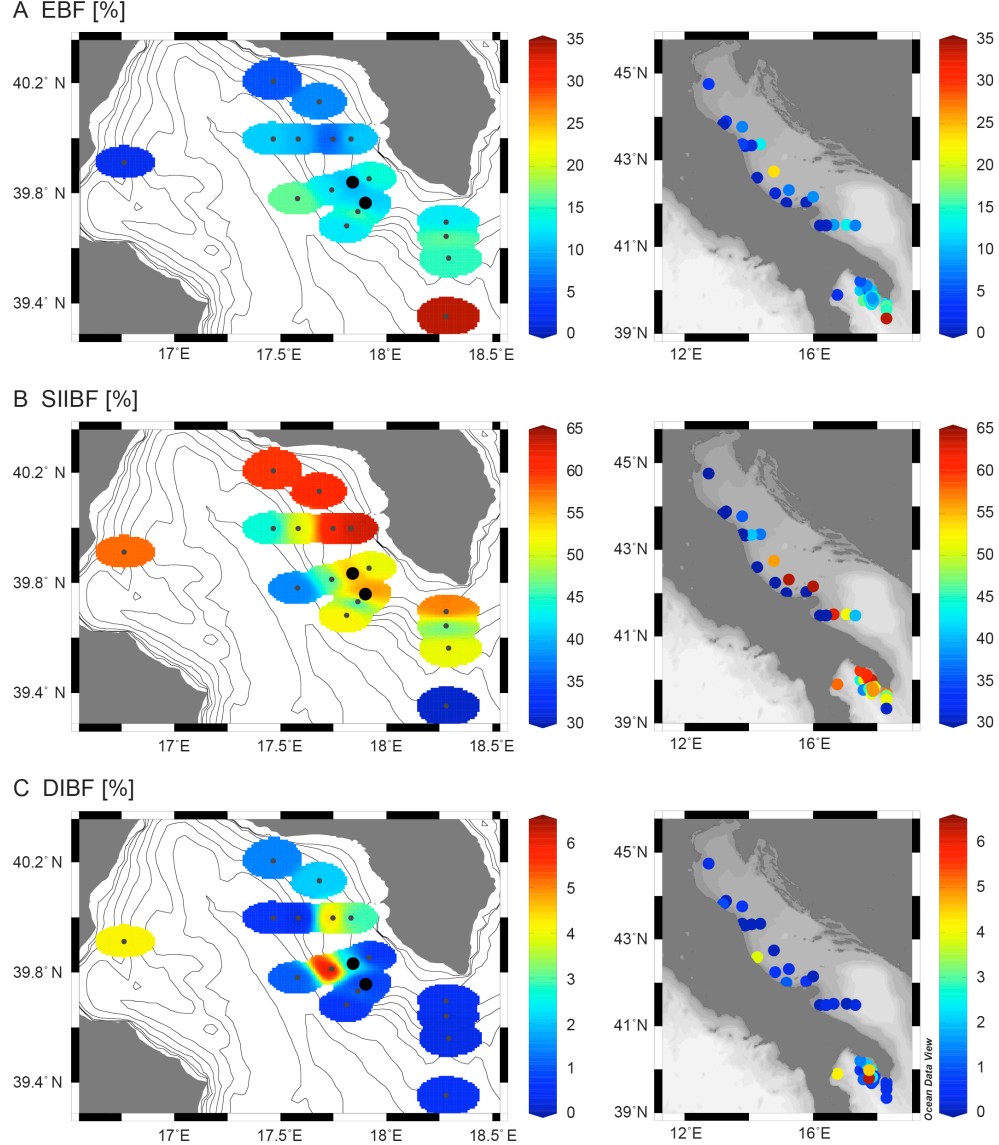

**Figure 6: Relative abundance of the main foraminiferal microhabitat groups (A-C) in surface sediment samples from the Gulf of Taranto and the western Adriatic Sea. Grey dots display the surface sediment samples, black bold dots show site 03 (lower right) and site 06 (upper left). EBF = epifaunal benthic foraminifera, SIIBF = shallow to intermediate infaunal benthic foraminifera, DIBF = deep infaunal benthic foraminifera.**



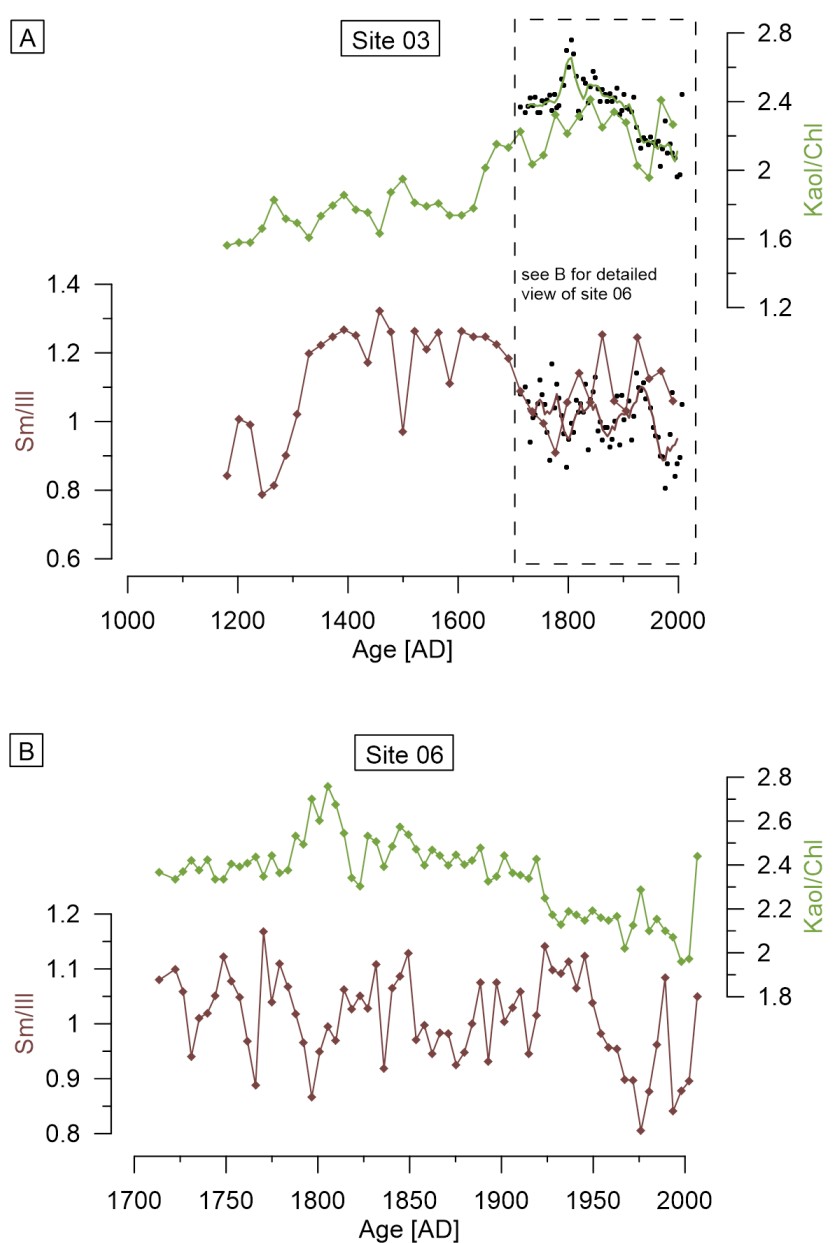

Figure 7: A) Clay mineral data of site 03 and site 06 with a detailed view of 06 in B) versus time. The kaolinite/chlorite ratio is shown in green, the smectite/illite ratio is shown in brown. Single black dots in A) depict data points from site 06, with the associated line presenting a five-point running average.




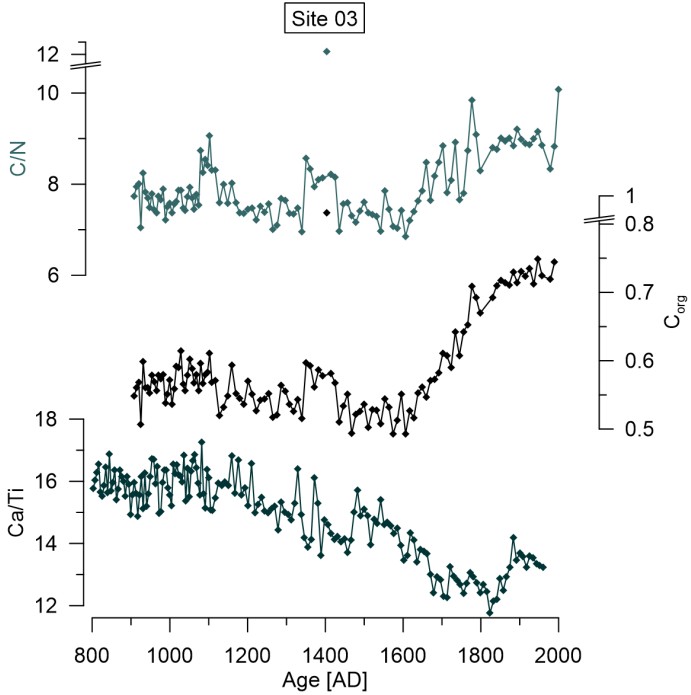

**Figure 8: C/N ratio, organic carbon content and Ca/Ti ratio of site 03 versus time.**





**Figure 9: Records for North Atlantic climate and solar activity compared to Italian environments for the last millennium. A)** Northern Hemisphere temperature anomaly (black) and strength of the Atlantic Multidecadal Oscillation (AMO) (gray) derived from average North Atlantic temperature anomalies (after Mann et al., 2009). **B)**





Relative abundance of shallow to intermediate infaunal benthic foraminifera (SIIBF) at site 03 (brown) and site 06 (orange dots; orange line displays the five point running average). C) C/N ratio of site 03 as indicator for the origin of organic matter. D) Smectite/ illite ratio of site 03 as indicator for Po river suspension load, with lower values suggesting higher Po River fluxes (dark brown). E) Atmospheric $\Delta^{14}C$ as a measure for solar activity (red; after

5    Reimer et al., 2004). F) Alkenone-based sea surface temperature (SST) record for the Gulf of Taranto (Versteegh et al., 2007). G) Ca/Ti ratio of site 03. H) Kaolinite/chlorite ratio of site 03 as an indicator for sediment input from the Apennines shown in light green. I) Reconstructed North Atlantic Oscillation (NAO) index (after Trouet et al., 2009). Orange triangles on the lower time axis mark the onset of the industrial revolution in Italy at around 1890 AD and the introduction of ammonia as fertilizer at 1913 AD. Dashed black lines show the onset and termination of alpine glacier

10   advances (Holzhauser et al., 2005). The Medieval Warm Period and Little Ice Age are also indicated (Lebreiro et al., 2006; Corona et al., 2010; Abrantes et al., 2005; Frisia et al., 2005).





**Figure 10: Comparison of records for North Atlantic climate, sunspot number, and Italian environments for the last three hundred years. A) Reconstructed North Atlantic Oscillation (NAO) index after Trouet et al. (2009, black line) and measured NAO index from NOAA (www.esrl.noaa.gov/psd/gcos_wgsp/Timeseries/NAO/) (red line). B) Smectite/ illite ratio of site 06 as indicator for Po river suspension load, with lower values suggesting higher Po river fluxes. Bold brown line represents a five point running average. C) Relative abundance of shallow to intermediate infaunal benthic**



foraminifera (SIIBF) at site 06 and thin line representing a five point running average. D) 21 point running average of the sunspot number from NOAA (https://www.ngdc.noaa.gov/stp/solar/ssndata.html). E) Kaolinite/ chlorite ratio of site 06 as indicator for sediment input from the Apennines. Bold light green line represents a five point running average. F) Relative abundance of epifaunal benthic foraminifera (EFB) of site 06. G) δ¹³C of *U.mediterranea* of site 06 as indicator for organic matter fluxes, with lower values suggesting higher organic matter fluxes. Bold dark green line represents a five point running average. Orange triangles on the lower time axis mark the unification of Italy at 1870 AD, the onset of the industrial revolution at around 1890 AD and the introduction of ammonia as fertilizer at 1913 AD. Dashed vertical line represents the termination of alpine glacier growth recorded by Holzhauser et al. (2005).

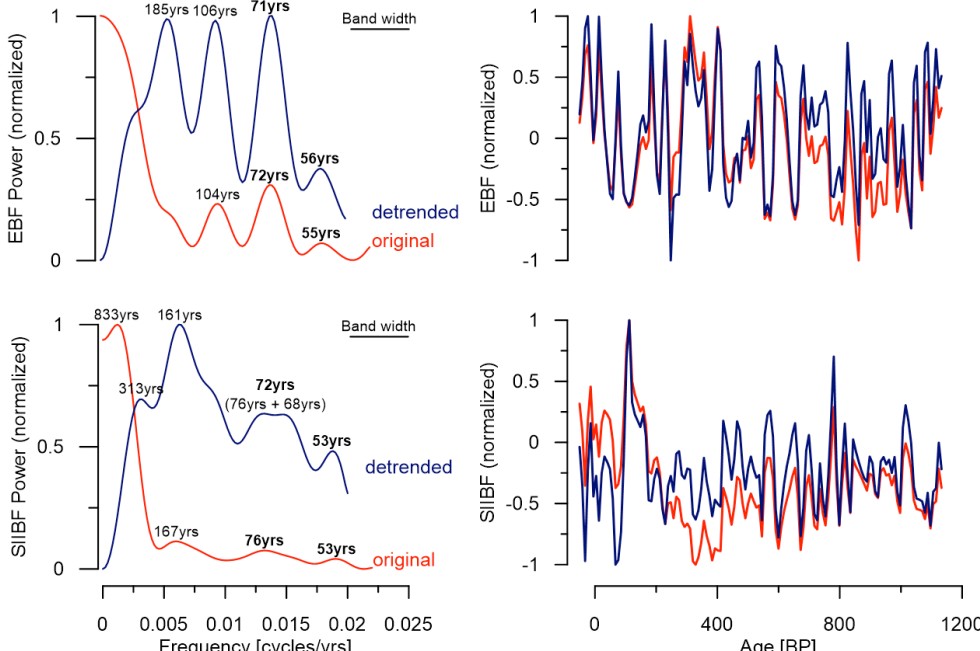

**Figure 11: Blackman-Tukey power spectra of the original (red) and detrended and normalized (blue) records of epifaunal and shallow infaunal benthic foraminifera of site 03. The foraminiferal data reveal periodic multi-decadal to centennial environmental variations.**