# Peer review of "Combined North Atlantic and anthropogenic forcing of changes in the marine environments in the Gulf of Taranto (Italy) during the last millennium"

_Climate of the Past, 2017_

## Referee Comment (RC1) · A Asioli (Referee) · 27 Dec 2017

The manuscript is of interest, as it focuses about the disentangling the natural and the anthropogenic forcing during the last millennium in a shelf environment off area presently densely populated (Mediterranean, Gulf of Taranto). The interpretation of the results of the study, carried out with a multiproxy approach, is quite complex, as expected, making the reading not always easy. The applied methods, although not new, are appropriated. Figures are all necessary and the quality is very good. Although I am not mother language, I had not problem to read the manuscript.

I have some comment, below listed, I wish the authors take into account:

Section: Environmental setting Page 3, lines 36-38: "The WAC (<37 psu; Lipizer et al., 2014) moves as a narrow coastal band along the western Adriatic margin and is further enriched in nutrients and sediment by smaller river systems of the Apennines before it reaches the Gulf of Taranto." The reference cited (Lipizer et al. 2014) does not seem to me to treat the path of the WAC beyond Otranto channel into the Gulf of Taranto, as well as other references cited in the same section, such as Budillon et al 2010, Artegiani et al. 1997, Turchetto et al., 2007. It is important that the authors provide the correct references for this topic, as the results of the study are linked to paleoclimatic scenarios of the northern Adriatic basin (Po river area). Moreover, the authors should report an appropriated reference (not Milligan and Cattaneo 2007) reporting the path of the Padane sediment flux aside (and not mixed with) the Appenine flux, as illustrated in Fig 1.

Page 3, lines 38-39 and page 4 lines 1-3: "The sediment supply from the southern Apennines is relatively small, but detectable near the coast through elevated smectite concentrations (Degrobbis et al., 1986; Tomadin 2000; Milligan and Cattaneo, 2007) (Fig. 1 A). The stronger Padane detrital matter flux dominated by illite (from rivers Po, Brenta, Adige, Reno) is seen in a parallel band further in the basin (e.g. Milligan and Cattaneo 2007; Tomadin 1979, 2000) (Fig. 1 A)." Milligan and Cattaneo (2007) is the introduction to a special issue dedicated to the sediment dynamic in the western Adriatic Sea. I recommend the authors to go through the papers of the special issue for a more complete presentation of the Adriatic sediment deposition and path (for instance Syvitski and Kettner regarding the sediment contribution of the Appenine rivers vs total Adriatic sediment load (ca. 50%))

Section Methods: Regarding the age-depth model, it seems from table 1 and fig. 2 that the modern time (2011 year of the core collection) is present. It is not clear to me if the authors assumed that the modern sediment is present or they based on other data (the sentence at page 5 line 5-6: "The core tops contain basically undisturbed surface sediments as indicated by the presence of an oxidized sediment layer" seems to me

not enough strong to support this assumption).

Page 6 lines 8-9 "The age model for GeoB15403-4 is based on 11 analyses of surface-dwelling planktonic foraminifera by the AMS 14C method". Please report the species of planktonic foraminifera picked up for the radiocarbon dates.

Page 6 lines 9-11: "Radiocarbon dates were converted to calendar years using the MARINE13 database (Reimer et al., 2013) with a Delta R value of 73±34". Please explain how the Delta R has been calculated.

Section 5 Discussion Page 9 lines 6-7. "Low Sm/Ill ratios therefore document a strong influence of the Padane flux. Minima occur around 1750 AD, 1470 AD and 1250 AD (Fig. 9 D). They coincide with high abundance of SIIBF (Fig. 9 B),...". I agree regarding the correspondence between 1750AD minima and abundance peak of SIIBF, but it is not much clear the correspondence between the other more recent Sm/Ill minima and SIIBF peaks, as the amplitude of the SIIBF fluctuations before 1750AD is not very high and it seems difficult to select the abundance peak corresponding to the Sm/Ill minima. The authors may trace correlation lines among the peaks.

Please check the text for typing errors (for instance Despart instead of Desprat page 2 line 30; Degrobbis instead of Degobbis page 3 line 39) while some reference in the list is not complete (for instance, Lipizer et al. 2014, Luterbacher et al., 2012, Rodolfi 1988)

I thank you very much for offering the opportunity to review this interesting paper.

December 27th 2017

---

## Referee Comment (RC2) · Anonymous Referee #2 · 25 Jan 2018

The paper submitted by Menke et alii is an attempt to demonstrate the influence of sediment released by the Italian rivers to the Sea and accumulated in the Gulf of Taranto (>1000km away from the Po River mouth). The authors examine decal to centennial variability of benthic ecosystems, depositional environments and bio-processes and relate them to natural (NAO, AMO) and anthropogenic forcing. Despite I enjoyed the data showed in figure 9, I cannot support the publication of the paper in it current state on Climate of the Past. Still, I cannot recommend the paper for publication for several reasons of which the most important are:

The Authors attempt to define the provenience of fine-sediment by looking at the rela-

tionship between illite and smectite. This methodology is based on earlier Tomadin's works which rely on old studies of sediment transport in the Adriatic basin. For instance, the Authors do not report new data on the source area of the Padane sector (i.e. the Padane sector is constituted by different catchments that are characterized by distinct sediment composition (see various papers by Eduardo Garzanti and Alessandro Amorosi in the Padane sector from '90s to now). In addition, the composition of the clay minerals does not reflect only the provenience of the sediment, especially in sediment that travelled such a long path. Finally, the key role of the interaction among different oceanic water masses has not been taken into account by Authors; this is true also for the different sediment facies that should reflect the depositional processes and may be examined in the available sediment cores. Moreover, recent documentation demonstrated that diagenesis play a crucial role even in modern sediment (e.g. see JHS Macquaker works on muddy deposits). Thus, the drivers of sediment transport and accumulation are different in nature and Authors seem to underestimating the role of each driver. Authors should study the sediment provenance by integrating different methods and by showing additional data.

As the Western Adriatic Current (WAC) has been defined in different ways from different authors (e.g. Artegiani et al., 1999 vs Poulain, 2001), Authors should state the evidence that the WAC is bringing sediment from the Adriatic Sea to the Gulf of Taranto. Perhaps, Authors should consider that: - from Artegiani et al. 1999, the WAC corresponds to the coastal amplification of the southeastward current on the mini-shelf and flows at water depth <20 m (the sediment cores showed in this work are at a water depth > 100 m). - Lipizer et al., 2014 suggest a sinking of the WAC along the slope of the western Adriatic margin and in the Otranto Strait (see their section 3.2.2) and they do not mention the Gulf of Taranto.

The Authors should provide references to works that document the path of the WAC in their study area. This is a fundamental point to address. How the sediment of the Po river reached the Gulf of Taranto? Authors should argue this concept with robust

references. What's the current that brings the sediment at the sites of the sediment cores sampled in the Gulf of Taranto? Indeed, as Authors reported in different sections the WAC flows along the Adriatic shelf. This is well documented from different works available from the bibliography. What remains unclear to the reader is the path of the WAC outside the Otranto Strait, based on previous works.

The concept that bands of sediment with Padane provenience and Apennine provenience travel in the Adriatic Sea on two "parallel highways" is an old concept based on earlier Tomadin's works. More recent works suggest a more complex oceanographic-sediment dispersion pattern (under the influence of water gyres and cascading see Trincardi et al., 2014). Moreover, based on his dataset, Tomadin doesn't take into account the distribution of the modern sediments and sampled deposits older than 5ky BP.

Readers would appreciate if Authors could add photos of the sediment cores (maybe in supplemental material?): any deformations of the cores due to gravity core sampling? If yes, how deformation has been taken into account? What is the recovery (%) of sediment cores?

The Authors should remind that they did not sample the depocenter, because the figures provided for sediment accumulation rates are telling the opposite (e.g. 28.3 cm/kyr that means a sediment accumulation rate of ca. 0.028 cm/yr... see Cattaneo et al., 2007 for maps of the modern depocenters and for the sediment accumulation rates (>1,5 cm/yr).

Part of the result section should be moved to methods (see highlighted sentences in the pdf).

Authors should discuss the Apennines sediment contribution on the light of Milliman and Syvitski, 1992, and Syvitski and Kettner, 2007.

Authors should avoid non-scientific language in many parts of the text (see comments

in the pdf).

Some references are missing from the reference list.

Fig. 1. I strongly suggest to avoid this very old concept of sediment transport, a lot of work has been done by different authors on the oceanographic regime and related sediment transport in the last decades. Authors should think about merging the two imagines in one maintaining the continental part from 1A and the marine part from 1B. Authors should provide references for the oceanographic regime in the Gulf of Taranto. Figs. 3 and 4. The resolution is good. Fig. 6. Authors should add isobaths values and a dotted line along the shelf-edge and should explicit that the endmembers of the color bars change from A to B to C. Figs. 9 and 10. I really like these figures! Maybe Authors can add lines to help the readers for correlation.

Here I am suggesting works on the Adriatic Sea dealing with the complexity of oceanographic mass water pathways and the sediment dispersal system: Oceanography:

-Benetazzo A, Bergamasco A, Bonaldo D, Falcieri FM, Sclavo M, Langone L, Carniel S (2014) Response of the Adriatic Sea to an intense cold air outbreak: dense water dynamics and wave-induced transport. Progr Oceanogr 128:115 – 138 -Bonaldo D, Benetazzo A, Bergamasco A, Campiani E, Foglini F, Sclavo M, Trincardi F, Carniel S (2015) Interactions among Adriatic continental margin morphology, deep circulation and bedform patterns. Mar Geol (in press). doi:10.1016/j.margeo.2015.09.012 -Artegiani A, Paschini E, Russo A, Bregant D, Raicich F, Pinardi N (1997) The Adriatic Sea general circulation. Part I: air – sea interactions and water mass structure. J Phys Oceanogr 27:1492 – 1514 -Artegiani A, Paschini E, Russo A, Bregant D, Raicich F, Pinardi N (1997) The Adriatic Sea general circulation. Part II: Baroclinic circulation structure. J Phys Oceanogr 27:1515 – 1532 Sediment dispersion and accumulation:

South Adriatic Sea, a work that shows also the interaction of current on the south Adriatic shelf.... -Pellegrini, C., Maselli, V., & Trincardi, F. (2016). Pliocene–Quaternary contourite depositional system along the south-western Adriatic margin: changes in

sedimentary stacking pattern and associated bottom currents. Geo-Marine Letters, 36(1), 67-79.

North and Central Adriatic Sea - Pellegrini, C., Maselli, V., Cattaneo, A., Piva, A., Ceregato, A., & Trincardi, F. (2015). Anatomy of a compound delta from the post-glacial transgressive record in the Adriatic Sea. Marine Geology, 362, 43-59. -Cattaneo, A., Trincardi, F., Asioli, A., Correggiari, A., 2007. The western Adriatic shelf clinoform: energy-limited bottomset. Cont. Shelf Res. 27, 506–525. Please follow the comments on the attached pdf (with tracking function). If the Authors cannot find support in the bibliography on the oceanographic regime and sediment dispersion they are suggesting as reference knowledge, they should consider switching the focus of the paper to a pure reconstruction of the past ca 1500 years. In this latter case, the Authors should make it clear the relevance of this new contribution in respect to the previous papers of the same working group. In case the Authors want to keep the same focus of their paper, I recommend Editors to send the manuscript to experts on the provenance of fine-grained sediments in marine environments. I am sorry about the negative conclusion, but hope that the comments can contribute to the improvement of the paper for a later submission. Anonymous January 25th, 2017

Please also note the supplement to this comment:
https://www.clim-past-discuss.net/cp-2017-139/cp-2017-139-RC2-supplement.pdf

**Supplement:**

**Combined North Atlantic and anthropogenic forcing of changes in the marine environments in the Gulf of Taranto (Italy) during the last millennium**

Valerie Menke1, Werner Ehrmann2, Yvonne Milker1, Swaantje Brzelinski1,3, Jürgen Möbius1,
Uwe Mikolajewicz4, Bernd Zolitschka5, Karin Zonneveld6, Kay Christian Emeis1, Gerhard Schmiedl1

[revised manuscript text omitted]
- 30 The clay mineral composition was analysed by X-ray diffraction (XRD) of the clay fraction. Samples were mounted as texturally oriented aggregates and solvated with ethylene-glycol vapour. The analyses were conducted on a Rigaku Miniflex system with CoKα radiation (30 kV, 15 mA). The samples were X-rayed in the range 3–40°20 with a step size of 0.02°20 and a measuring time of 2 s/step. We additionally analysed the range 27.5–30.6°20 with a step size of 0.001°20 and a measuring time of 4 s/step in order to better resolve the (002)
- 35 peak of kaolinite and the (004) peak of chlorite. The individual clay minerals were identified by their basal reflections. For semi-quantitative evaluations of the clay mineral assemblages we used 
[revised manuscript text omitted]

---

## Referee Comment (RC3) · Anonymous Referee #3 · 18 Feb 2018

Dear Natascha Töpfer Copernicus Publications Editorial Support

I hereby you receive my report on the MS "Combined North Atlantic and anthropogenic forcing of changes in the marine environments in the Gulf of Taranto (Italy) during the last millennium" by Menke et al The authors analysed benthic foraminifera and clay mineral data providing information on natural and the anthropogenic forcing during the last millennium in a shelf area of Taranto Gulf. This area has been intensively studied (about 19 scientific articles) by several authors (from Cini-Castagnoli et al. 1989 to Goudeau et al. 2014) using different tools. In all these scientific articles, it is evident the extreme high sedimentation rate of this area mainly over the last millennium,

but it also evident the occurrence of several problem in terms of chronology (see the high-error bars for AMS14C dating, Grauel et al 2013; or Versteegh et al 2007 where it is impossible to understand the chronology and where the references reported for the chronology have no data). In most of these articles is reported a more or less constant sed-rate over the last two millennia. It is surprising to document constant Sed.Rate in a shelf area with a dynamic marine environment. Due to the water depth of the marine sediments recovered in the cores, when the authors described the sediment they have to use the term hemipelagic sediment and not "homogeneous nannofossil-rich mud". In addition, due to the goal of the manuscript it is necessary to be more precise in the description of the sediment recovered in the cores. Are there other biota with benthic foraminifera? I know that in this area, there are also layers with ostracods and in some case, there are also pteropods. Are there evidence of this biota? In my opinion, the first problem is related for the last two centuries. There are no radionuclides data (210Pb and 137Cs). Did the authors consider the radionuclides data published in Grauel et al (2013) to create the age model for the last centuries? Without radionuclides data, the authors cannot produce an age model for the last two centuries (i.e. last 150 years) to demonstrate the continuous and normal sedimentation rate in the upper most part of the core. The second problem is associated to the occurrence of a tephra layer related to Lipari Eruption. The authors reported as reference for this tephra, a manuscript submitted as Menke et al.. I do not think that the authors can use this tephra as tie-point, reporting as reference a manuscript under review. In addition, the authors refer this tephra layer to Lipari eruption, but this eruption is related to Monte Pilato eruption phases (see Forni et al., 2013, Geological Society of London). This Monte Pilato volcanic phase spans from 729 to 1220 AD (see Forni et al., 2013, Geological Society of London). Why did the authors decide to associate this tephra layer to an age of 1174 yr BP (776 yr AD) that corresponds (or it is very close) to the age associated to the beginning of this eruption phase? Maybe it is wright, but if this reconstruction is based on data reported in the manuscript Menke et al. under review, I think that the authors have to improve the present version of the manuscript. Alternatively, they have to exclude this tephra from the present age model. The authors reported from line 5 to line 10 (pag 5) information concerning the undisturbed sediment in the uppermost part of the records. Without radionuclides and proxy of porosity, it is not possible to make this assumption. There no evidence to support this assumption. In table 1, it is necessary to specify if the authors run AMS14C on mix of planktonic foraminifera or on single species. In addition, it is important to indicate the thickness of the sample used for each AMS14C dating. This information is important to analyse the propagation of errors. In your age-depth profile (Fig. 2) it is important to show the propagation of errors. Because of the authors have no radionuclides data, the propagation of errors cannot be extended to top core. Please it is necessary to show the algorithm used to create the age model. Spectral analyses: I would like to suggest to use the "Intrinsic Mode Functions" (IMF) (Huang et al., 1998) to analyse the signal and to run wavelet analysis on selected IMF component. This is the correct approach when you analysis records for the last millennia. Only with this approach you can identify the stable frequency associated to your proxy and if it is continuous present along the whole study record. The single spectrum reported in figure 11 represents a mean value within the record, so that it is not representative of a possible forcing. Concerning the recently scientific literature focused on the shallow water environment, the authors did not consider in this manuscript several articles (Ferraro et al., 2012; Vallefuoco et al., 2012; Lirer et al., 2014; Margaritelli et al., 2016; Di Rita et al., 2018; Oldfield et al. 2003; Bonomo et al., 2016; Di Bella et al., 2014). These articles in my opinion offer some important information for the submitted manuscript. Ferraro et al. (2012) and Di Bella et al. (2014) for benthic foraminifera, Bonomo et al. (2016) and Di Rita et al. (2018) relation between NAO and runoff/alboreal pollen, Oldfield et al. (2003) with low resolution concerning benthic foraminifera, etc. ….. Are there specific reasons for this choice? Concerning the NAO index, I think that the article Brunetti et al (2002) focused on the winter precipitation in Italy modulated by NAO, has to be take in account. In addition, the NAO forcing has been shown also in other fossil marine sedimentary archives by several authors (Chen et al., 2011; Nieto-Moreno et al., 2013; Goudeau et al., 2015; Jalali et al., 2015).

Why the authors did not consider these references from Mediterranean area? In Figure 3A, the authors compared SIBF signals in the two records, but as documented in the figure for the last two centuries, the two curves have an antithetic pattern. The same framework, maybe less pronounced, is also shown for SIIBF signals. In my opinion, in the manuscript the explanation reported for this discrepancy in the last two centuries is not scientific supported. In my opinion, without radionuclides dating this problem cannot be solved. Again, I would like to suggest to the authors to plot the distribution patterns of benthic foraminifera per gr of sediment to understand or to try to interpret correctly this discrepancy. In my opinion, the differences in benthic assemblage reported for both study sites in figure 6, is not so evident. In addition, without a detailed high-resolution morphobatimetry of the study area it is not possible to propose this type of interpretation. The authors have to focus as follows: 1) on chronology of the last two centuries and on the determination of propagation of errors, 2) on the interpretation of benthic foraminifera per gr of sediment and not in percentages, due to the target of the manuscript. This approach could help to interpret the benthic data vs the target of this manuscript. 3) I think that the authors have to take in account also the several dams build along the rivers of the Adriatic Sea. These constructions changed significantly the sediment outflow in Adriatic Sea. 4) I would like to suggest to see also the contribution of Ofanto river. 5) It is necessary to improve the spectral and wavelet analysis carried out on the proxies. 6) If is necessary to filter each frequency and compared these with internal or external forcing. 7) Due to the target of the manuscript it is necessary to compare the study records (biotic or abiotic proxies) with proxy of river discharge My overall conclusion is that the manuscript is properly constructed and is suitable for the journal but unfortunately, it needs major revision.

---

## Author Comment (AC1) · 14 Mar 2018

Response to Reviewer Comment 1

We would like to thank Alessandra Asioli for the helpful comments and suggestions that will help to improve the manuscript. Below we added our point to point reply to all comments. Referee comments are displayed in italic font, our response is written in normal font.

*1) Section: Environmental setting Page 3, lines 36-38: "The WAC (<37 psu; Lipizer et al., 2014) moves as a narrow coastal band along the western Adriatic margin and is further enriched in nutrients and sediment by smaller river systems of the Apennines before it reaches the Gulf of Taranto." The reference cited (Lipizer et al. 2014) does not seem to me to treat the path of the WAC beyond Otranto channel into the Gulf of Taranto, as well as other references cited in the same section, such as Budillon et al 2010, Artegiani et al. 1997, Turchetto et al., 2007. It is important that the authors provide the correct references for this topic, as the results of the study are linked to paleoclimatic scenarios of the northern Adriatic basin (Po river area). Moreover, the authors should report an appropriated reference (not Milligan and Cattaneo 2007) reporting the path of the Padane sediment flux aside (and not mixed with) the Appenine flux, as illustrated in Fig 1.*

We agree with the reviewer that in the settings section we mostly cite publications on the oceanography of the water currents up to the Strait of Otranto. To improve this section and underline our argumentation we will for example include studies of: Goudeau et al. (2014) (finding evidence that the dominant provenance of Gallipoli Shelf sediments originates from the western Adriatic mud belt, transported by the WAC), Grauel et al. (2013 a, b) (finding evidence that relatively nutrient-rich and fresher waters of the WAC influence the isotopic composition of surface-dwelling planktonic foraminifera on multi-decadal time scales) and Zonneveld et al. (2012) (finding a correlation between Po River discharge and the accumulation and relative abundance of dinoflagellate cysts in samples from the Gulf of Taranto and the Gulf of Manfredonia).
We agree with the reviewer that our phrasing suggests a strict separation between the Padane and Apennine sediment flux and we don't emphasize enough, that there is a mixing between the two. We will rephrase the corresponding section in the introduction and modify Figure 1 in the manuscript as suggested by reviewer 2. Furthermore, we generated a new record of clay mineral data using surface samples from the Adriatic Sea and the Gulf of Taranto to show that there is a spatial gradient in Padane and Apennine generated material (see Response to Reviewer Comment (RRC) 2 Figure 1). We will add a corresponding figure (in the style of Figure 6 of the current manuscript) and explanation to the main body of the text and rephrase respective sections in the discussion accordingly.

*2) Page 3, lines 38-39 and page 4 lines 1-3: "The sediment supply from the southern Apennines is relatively small, but detectable near the coast through elevated Smectite concentrations (Degrobbis et al., 1986; Tomadin 2000; Milligan and Cattaneo, 2007) (Fig. 1 A). The stronger Padane detrital matter flux dominated by illite (from rivers Po, Brenta, Adige, Reno) is seen in a parallel band further in the basin (e.g. Milligan and Cattaneo 2007; Tomadin 1979, 2000) (Fig. 1 A)." Milligan and Cattaneo (2007) is the introduction to a special issue dedicated to the sediment dynamic in the western Adriatic Sea. I recommend the authors to go through the papers of the special issue for a more complete presentation of the Adriatic sediment deposition and path (for instance Syvitski and Kettner regarding the sediment contribution of the Appenine rivers vs total Adriatic sediment load (ca. 50%)).*

We agree with the reviewer that our description of the depositional environment could be improved. We will include studies of for example Syvitski and Kettner (2007), Milliman and

Syvitski (1992) and Amorosi et al. (2016) and elaborate more on the northern Apennine sediment contribution to the Po River sediment flux and the central Apennine sediment contribution to the Adriatic Sea sediment load.

*3) Section Methods: Regarding the age-depth model, it seems from table 1 and fig. 2 that the modern time (2011 year of the core collection) is present. It is not clear to me if the authors assumed that the modern sediment is present or they based on other data (the sentence at page 5 line 5-6: "The core tops contain basically undisturbed surface sediments as indicated by the presence of an oxidized sediment layer" seems to me not enough strong to support this assumption).*

The maximum depth of oxygen penetration into the surface sediment of gravity core GeoB 15403-4 was identified by distinct color changes at 6 cm. The oxygen penetration depth in a multicore from the same station (GeoB15403-6) was identified at 8cm sediment depth. Furthermore, when the core was opened and sampled, the sediment surface appeared undisturbed with no indications of compression or sediment loss as seen on core pictures. Therefore we argue, that the core top represents the year 2011 when the core was retrieved. To support this, we correlated Ca/Ti data from XRF core scanning to the adjacent sediment core DP30, which contains modern sediments at the core top as proven by $^{210}$Pb dating (Goudeau et al., 2014) (see RRC3, Figure 1). Although accumulation rates at the two sites are different, the close correspondence of the Ca/Ti records (further validated by the available AMS $^{14}$C ages) suggests undisturbed records and confirms the reliability of our age model. The presence of an undisturbed sediment surface in GeoB15403-4 is further supported by the continuation of the Ca/Ti curve at site GeoB15403-4 since it was sampled few years after core DP30. Explanations will be added to the manuscript appropriately.

*4) Page 6 lines 8-9 "The age model for GeoB15403-4 is based on 11 analyses of surface dwelling planktonic foraminifera by the AMS 14C method". Please report the species of planktonic foraminifera picked up for the radiocarbon dates.*

We selected shells of surface dwelling planktonic foraminifera, preferentially *Globigerinoides ruber* (variety pink and white), *Globigerinoides conglobatus*, *Globigerina bulloides*, *Orbulina universa* and *Globigerinella siphonifera*. We avoided shells of subsurface-dwelling taxa such as *Globorotalia inflata, Globorotalia truncatulinoides,* and *Neogloboquadrina dutertrei.* This information will be added to the methods section.

*5) Page 6 lines 9-11: "Radiocarbon dates were converted to calendar years using the MARINE13 database (Reimer et al., 2013) with a Delta R value of 73_34". Please explain how the Delta R has been calculated.*

Delta R was calculated based on a weighted mean between the adjacent stations MapNo 234 (Lon.: 40.83; Lat.: 18.33) and MapNo 238 (Lon.: 42.5; Lat.:17) derived from the marine reservoir correction database MarineChrono (www.calib.org/marine/). We will include this information in the methods section of the manuscript.

*6) Section 5 Discussion Page 9 lines 6-7. "Low Sm/Ill ratios therefore document a strong influence of the Padane flux. Minima occur around 1750 AD, 1470 AD and 1250 AD (Fig. 9 D). They coincide with high abundance of SIIBF (Fig. 9 B). I agree regarding the correspondence between 1750AD minima and abundance peak of SIIBF, but it is not much clear the correspondence between the other more recent Sm/Ill minima and SIIBF peaks, as*

*the amplitude of the SIIBF fluctuations before 1750AD is not very high and it seems difficult to select the abundance peak corresponding to the Sm/III minima. The authors may trace correlation lines among the peaks.*

We will consider this issue in the revised version, either by modification of Figure 9 to show the correlation more clearly or by rephrasing the description accordingly.

*7) Please check the text for typing errors (for instance Despart instead of Desprat page 2 line 30; Degrobbis instead of Degobbis page 3 line 39) while some reference in the list is not complete (for instance, Lipizer et al. 2014, Luterbacher et al., 2012, Rodolfi 1988).*

All references will be double checked and typing errors will be corrected.

Cited References:

Amorosi, A., Maselli, V., and Trincardi, F., 2016, Onshore to offshore anatomy of a late Quaternary source-to-sink system (Po Plain–Adriatic Sea, Italy): Earth-Science Reviews, v. 153, p. 212-237.

Goudeau, M. L. S., Grauel, A. L., Tessarolo, C., Leider, A., Chen, L., Bernasconi, S. M., Versteegh, G. J. M., Zonneveld, K. A. F., Boer, W., Alonso-Hernandez, C. M., and De Lange, G. J., 2014, The Glacial-Interglacial transition and Holocene environmental changes in sediments from the Gulf of Taranto, central Mediterranean: Marine Geology, v. 348, p. 88-102.

Grauel, A.-L., Goudeau, M.-L. S., de Lange, G. J., and Bernasconi, S. M., 2013 a, Climate of the past 2500 years in the Gulf of Taranto, central Mediterranean Sea: A high-resolution climate reconstruction based on $\delta 18O$ and $\delta 13C$ of Globigerinoides ruber (white): The Holocene, v. 23, no. 10, p. 1440-1446.

Grauel, A. L., Leider, A., Goudeau, M. L. S., Muller, I. A., Bernasconi, S. M., Hinrichs, K. U., de Lange, G. J., Zonneveld, K. A. F., and Versteegh, G. J. M., 2013 b, What do SST proxies really tell us? A high-resolution multiproxy (U-37(K '), TEX86H and foraminifera delta O-18) study in the Gulf of Taranto, central Mediterranean Sea: Quaternary Science Reviews, v. 73, p. 115-131.

Milliman, J. D., and Syvitski, J. P., 1992, Geomorphic/tectonic control of sediment discharge to the ocean: the importance of small mountainous rivers: The Journal of Geology, v. 100, no. 5, p. 525-544.

Syvitski, J., and Kettner, A. J., 2007, On the flux of water and sediment into the Northern Adriatic Sea: Continental Shelf Research, v. 27, no. 3-4, p. 296-308.

Zonneveld, K. A., Chen, L., Elshanawany, R., Fischer, H. W., Hoins, M., Ibrahim, M. I., Pittauerova, D., and Versteegh, G. J., 2012, The use of dinoflagellate cysts to separate human-induced from natural variability in the trophic state of the Po River discharge plume over the last two centuries: Marine pollution bulletin, v. 64, no. 1, p. 114-132.

---

## Author Comment (AC2) · 14 Mar 2018

Response to Reviewer Comment 2

We would like to thank the anonymous referee 2 for the detailed comments and suggestions that will help to improve the manuscript substantially. Below we added our point to point reply to all comments. Referee comments are displayed in italic font, our response is written in normal font.

*1) The Authors attempt to define the provenience of fine-sediment by looking at the relationship between illite and smectite. This methodology is based on earlier Tomadin's works which rely on old studies of sediment transport in the Adriatic basin. For instance, the Authors do not report new data on the source area of the Padane sector (i.e. the Padane sector is constituted by different catchments that are characterized by distinct sediment composition (see various papers by Eduardo Garzanti and Alessandro Amorosi in the Padane sector from '90s to now). In addition, the composition of the clay minerals does not reflect only the provenience of the sediment, especially in sediment that travelled such a long path. Finally, the key role of the interaction among different oceanic water masses has not been taken into account by Authors; this is true also for the different sediment facies that should reflect the depositional processes and may be examined in the available sediment cores. Moreover, recent documentation demonstrated that diagenesis play a crucial role even in modern sediment (e.g. see JHS Macquaker works on muddy deposits). Thus, the drivers of sediment transport and accumulation are different in nature and Authors seem to underestimating the role of each driver. Authors should study the sediment provenance by integrating different methods and by showing additional data.*

We generated an additional dataset on the clay mineral distribution in the seafloor surface sediments of the Gulf of Taranto and the western Adriatic Sea to further assess the provenance of the different clay minerals and spatial gradients in clay mineral dispersal. Response to Reviewer Comment (RRC) 2 Figure 1 shows first results. We will present these new data in form of a new figure showing clay mineral ratios in the style of Figure 6 of the current manuscript. This will also allow us to further elaborate on the relation between different water masses and the clay mineral transport and distribution and on the integrated signal of the different sediment sources.
Diagenesis of clay minerals normally requires increased temperature and pressure and therefore is confined to deeper and older parts of marine sedimentary sequences (see also Brindley and Brown, 1980, Chamley, 1989, Meunier, 2005). Also our previous studies on modern, Holocene and Late Pleistocene clay minerals in various parts of the eastern Mediterranean Sea (e.g., Ehrmann et al., 2007 a,b; Hamann et al., 2009; Ehrmann et al., 2013, 2016, 2017) did not give any indication of a diagenetic alteration but suggest that the clay mineral distribution is mainly controlled by sediment provenance and dispersal.

[Figure]

**RRC2 Figure 1:** Map showing first results from the clay mineral analysis of surface samples from the Gulf of Taranto and the western Adriatic Sea. The data show an on-offshore gradient and trace the near-coastal transport of suspension from the Adriatic Sea into the Gulf of Taranto. In the revised manuscript, results will be presented in form of clay mineral ratios as also applied to the sediment core.

*2) As the Western Adriatic Current (WAC) has been defined in different ways from different authors (e.g. Artegiani et al., 1999 vs Poulain, 2001), Authors should state the evidence that the WAC is bringing sediment from the Adriatic Sea to the Gulf of Taranto. Perhaps, Authors should consider that: - from Artegiani et al. 1999, the WAC corresponds to the coastal amplification of the southeastward current on the mini-shelf and flows at water depth 100 m). - Lipizer et al., 2014 suggest a sinking of the WAC along the slope of the western Adriatic margin and in the Otranto Strait (see their section 3.2.2) and they do not mention the Gulf of Taranto. The Authors should provide references to works that document the path of the WAC in their study area. This is a fundamental point to address. How the sediment of the Po river reached the Gulf of Taranto? Authors should argue this concept with robust paper references. What's the current that brings the sediment at the sites of the sediment cores sampled in the Gulf of Taranto? Indeed, as Authors reported in different sections the WAC flows along the Adriatic shelf. This is well documented from different works available from the bibliography. What remains unclear to the reader is the path of the WAC outside the Otranto Strait, based on previous works.*

We agree with the reviewer that in the settings section we mostly cited publications on the oceanography of the water currents up to the Strait of Otranto. To improve this section and underline our argumentation we will for example include studies of: Goudeau et al. (2014) (finding evidence that the dominant provenance of Gallipoli Shelf sediments originates from the western Adriatic mud belt, transported by the WAC), Grauel et al. (2013 a, b) (finding evidence that relatively nutrient-rich and fresher waters of the WAC influence the isotopic composition of surface dwelling planktonic foraminifera on multi-decadal time scales) and Zonneveld et al. (2012) (finding a correlation between Po River discharge and the accumulation and relative abundance of dinoflagellate cysts in samples from the Gulf of Taranto and the Gulf of Manfredonia). Furthermore we generated a new clay mineral record of surface samples from the Gulf of Taranto and the Adriatic Sea (RRC2 Figure 1), which traces the near-coastal suspension transport from the Adriatic Sea into the Gulf of Taranto. We will add a corresponding figure (in the style of Figure 6 of the current manuscript version)

displaying clay mineral ratios to assess the modern clay mineral distribution. This will allow for a further elaboration on clay mineral dispersal pathways.

*3) The concept that bands of sediment with Padane provenience and Apennine provenience travel in the Adriatic Sea on two "parallel highways" is an old concept based on earlier Tomadin's works. More recent works suggest a more complex oceanographic sediment dispersion pattern (under the influence of water gyres and cascading see Trincardi et al., 2014). Moreover, based on his dataset, Tomadin doesn't take into account the distribution of the modern sediments and sampled deposits older than 5ky BP.*

We agree with the reviewer that our phrasing suggests a strict separation between the Padane and Apennine sediment flux and we did not emphasize the mixing between the different sources sufficiently. We will rephrase the corresponding section in the introduction and refer to relevant literature.
The cited study of Tomadin (2000) did not sample deposits older than 5 ky BP but investigated surface sediments as stated by the author: "More than 300 bottom samples, collected by van Veen grab samplers, box-corers and piston corers during numerous cruises, have been considered for the present investigation. To avoid the comparison of materials of different age, thin layers of sediments deposited on the sea floor were carefully sampled."
Even though local currents and gyres play an important role in sediment dispersal, the general concept of a spatial gradient in Po- and Apennine-generated material is convincing in our opinion. To further validate this, we generated a new clay mineral record of surface samples from the Gulf of Taranto and the western Adriatic Sea (RRC2 Figure 1 provides first results). We will add one or two additional figures and a corresponding paragraph to better assess the modern clay mineral distribution and dispersal pathways in the study area.

*4) Readers would appreciate if Authors could add photos of the sediment cores (maybe in supplemental material?): any deformations of the cores due to gravity core sampling? If yes, how deformation has been taken into account? What is the recovery (%) of sediment cores?*

The sediment cores display an undisturbed sediment surface and no signs of deformation or sediment loss are visible. A picture of the gravity core will be added in the supplementary material.

*5) The Authors should remind that they did not sample the depocenter, because the figures provided for sediment accumulation rates are telling the opposite (e.g. 28.3 cm/kyr that means a sediment accumulation rate of ca. 0.028 cm/yr... see Cattaneo et al., 2007 for maps of the modern depocenters and for the sediment accumulation rates (>1,5 cm/yr).*

The sedimentation rates at Site 06 are relatively high (approximately 1 mm/yr) considering the core location at 214m water depth in the Gulf of Taranto far away from a large river mouth or the main Apennine sediment sources. Mesotrophic to eutrophic conditions displayed by the abundance of SIIBF suggest high accumulation rates of organic material hence documenting a depocenter in the northeastern part of the Gulf of Taranto. We are aware of the fact that there are areas in the western Adriatic Sea closer to the Apennine and Po sediment sources with far higher sediment accumulation rates. Nevertheless, we would like to retain this wording considering our study area in the Gulf of Taranto. The sedimentation rates for Site 03 are lower when compared to Site 06 and we point out that this is most likely due to its location at the margin of the depocenter. In the revised manuscript, however, we will refer to the study of Cattaneo et al. (2007) in order to put the sediment accumulation in the Gulf of Taranto into a regional perspective.

*6) Part of the result section should be moved to methods (see highlighted sentences in the pdf).*

We will shift the definition of the microhabitat groups (e.i. SIIBF, EBF) and the related discussion into the methods chapter.

*7) Authors should discuss the Apennines sediment contribution on the light of Milliman and Syvitski, 1992, and Syvitski and Kettner, 2007.*

We agree with the reviewer that the manuscript would benefit from a more detailed description of the depositional environment. We will include the studies of, for example, Syvitski and Kettner (2007), Milliman and Syvitski (1992) and Amorosi et al. (2016) and elaborate more on the northern Apennine sediment contribution to the Po River sediment flux and the central Apennine sediment contribution to the total sediment load of the Adriatic Sea.

*8) Authors should avoid non-scientific language in many parts of the text (see comments in the pdf).*

The use of language will be checked throughout the whole manuscript and modified appropriately.

*9) Some references are missing from the reference list.*

All references will be double checked.

*10) Fig. 1. I strongly suggest to avoid this very old concept of sediment transport, a lot of work has been done by different authors on the oceanographic regime and related sediment transport in the last decades. Authors should think about merging the two imagines in one maintaining the continental part from 1A and the marine part from 1B. Authors should provide references for the oceanographic regime in the Gulf of Taranto.*

We agree with the reviewer and will combine Figure 1 A and B into one figure merging the terrestrial part of Figure 1 A with the marine part of Figure 1 B and will avoid illustration of a strict separation between a Padane and an Apennine flux. Furthermore, we suggest to modify the current division of the Apennines according to the different Apennine sediment contribution into the Adriatic Sea (eastern Apennine Rivers: $32.5 \times 10^6$ t yr$^{-1}$, Apennine Rivers south of Gargano: $1.5 \times 10^6$ t yr$^{-1}$).

*11) Figs. 3 and 4. The resolution is good. Fig. 6. Authors should add isobaths values and a dotted line along the shelf-edge and should explicit that the endmembers of the color bars change from A to B to C. Figs. 9 and 10. I really like these figures! Maybe Authors can add lines to help the readers for correlation.*

We will add information on the different color bars and display the values for the depth of the isobaths.

*12) Here I am suggesting works on the Adriatic Sea dealing with the complexity of oceanographic mass water pathways and the sediment dispersal system:*

We appreciate the recommended literature and will consider the publications in appropriate sections of the manuscript.

Cited References

Amorosi, A., Maselli, V., and Trincardi, F., 2016, Onshore to offshore anatomy of a late Quaternary source-to-sink system (Po Plain–Adriatic Sea, Italy): Earth-Science Reviews, v. 153, p. 212-237.

Brindley, G.W., Brown, G. (Editors), 1980. Crystal Structures of Clay Minerals and their X-Ray Identification. Mineral. Soc. Monogr., 5: 495 pp.

Chamley, H. (1989) Clay Sedimentology. Berlin, Heidelberg, New York: Springer, 623 pp.

Cattaneo, A., Trincardi, F., Asioli, A., and Correggiari, A., 2007, The Western Adriatic shelf clinoform: energy-limited bottomset: Continental Shelf Research, v. 27, no. 3-4, p. 506-525.

Ehrmann, W., Schmiedl, G., Hamann, Y., and Kuhnt, T., 2007a, Distribution of clay minerals in surface sediments of the Aegean Sea: a compilation: International Journal of Earth Sciences, v. 96, no. 4, p. 769.

Ehrmann, W., Schmiedl, G., Hamann, Y., Kuhnt, T., Hemleben, C., and Siebel, W., 2007b, Clay minerals in late glacial and Holocene sediments of the northern and southern Aegean Sea: Palaeogeography, Palaeoclimatology, Palaeoecology, v. 249, no. 1, p. 36-57.

Ehrmann, W., Schmiedl, G., Beuscher, S., and Krüger, S., 2017, Intensity of African Humid Periods estimated from Saharan dust fluxes: PloS one, v. 12, no. 1, p. e0170989.

Ehrmann, W., Schmiedl, G., Seidel, M., Krüger, S., and Schulz, H., 2016, A distal 140 kyr sediment record of Nile discharge and East African monsoon variability: Climate of the Past, v. 12, no. 3, p. 713-727.

Ehrmann, W., Seidel, M., and Schmiedl, G., 2013, Dynamics of Late Quaternary North African humid periods documented in the clay mineral record of central Aegean Sea sediments: Global and planetary change, v. 107, p. 186-195.

Goudeau, M. L. S., Grauel, A. L., Tessarolo, C., Leider, A., Chen, L., Bernasconi, S. M., Versteegh, G. J. M., Zonneveld, K. A. F., Boer, W., Alonso-Hernandez, C. M., and De Lange, G. J., 2014, The Glacial-Interglacial transition and Holocene environmental changes in sediments from the Gulf of Taranto, central Mediterranean: Marine Geology, v. 348, p. 88-102.

Grauel, A.-L., Goudeau, M.-L. S., de Lange, G. J., and Bernasconi, S. M., 2013a, Climate of the past 2500 years in the Gulf of Taranto, central Mediterranean Sea: A high-resolution climate reconstruction based on $\delta 18O$ and $\delta 13C$ of Globigerinoides ruber (white): The Holocene, v. 23, no. 10, p. 1440-1446.

Grauel, A. L., Leider, A., Goudeau, M. L. S., Muller, I. A., Bernasconi, S. M., Hinrichs, K. U., de Lange, G. J., Zonneveld, K. A. F., and Versteegh, G. J. M., 2013b, What do SST proxies really tell us? A high-resolution multiproxy (U-37(K '), TEX86H and foraminifera delta O-18) study in the Gulf of Taranto, central Mediterranean Sea: Quaternary Science Reviews, v. 73, p. 115-131.

Hamann, Y., Ehrmann, W., Schmiedl, G., and Kuhnt, T., 2009, Modern and late Quaternary clay mineral distribution in the area of the SE Mediterranean Sea: Quaternary Research, v. 71, no. 3, p. 453-464.

Meunier, A., 2005. Clays. Springer, Berlin, Heidelberg, New York, 472 pp.

Milliman, J. D., and Syvitski, J. P., 1992, Geomorphic/tectonic control of sediment discharge to the ocean: the importance of small mountainous rivers: The Journal of Geology,

v. 100, no. 5, p. 525-544.

Syvitski, J., and Kettner, A. J., 2007, On the flux of water and sediment into the Northern Adriatic Sea: Continental Shelf Research, v. 27, no. 3-4, p. 296-308.

---

## Author Comment (AC3) · 14 Mar 2018

Response to Reviewer Comment 3

We would like to thank the anonymous referee 3 for the detailed comments and suggestions that will help to improve the manuscript substantially. Below we respond, point by point, to all comments. Referee comments are displayed in italic font, our response is written in normal font.

*1) Due to the water depth of the marine sediments recovered in the cores, when the authors described the sediment they have to use the term hemipelagic sediment and not "homogeneous nannofossil-rich mud". In addition, due to the goal of the manuscript it is necessary to be more precise in the description of the sediment recovered in the cores. Are there other biota with benthic foraminifera? I know that in this area, there are also layers with ostracods and in some case, there are also pteropods. Are there evidence of this biota?*

We will add the term "hemipelagic sediment" for the general characterization of the sediments. The studied core intervals are rather homogenous. Lithologically, the sediment can be correctly classified as nannofossil- and foraminifer-bearing mud. The sediment further contains traces of pteropods, ostracods, echinoid spines and other microfossils, which are not concentrated in layers but are more or less homogenously distributed over the entire interval studied. We will modify the text accordingly.

*2) In my opinion, the first problem is related for the last two centuries. There are no radionuclides data (210Pb and 137Cs). Did the authors consider the radionuclides data published in Grauel et al (2013) to create the age model for the last centuries? Without radionuclides data, the authors cannot produce an age model for the last two centuries (i.e. last 150 years) to demonstrate the continuous and normal sedimentation rate in the upper most part of the core.*

The maximum depth of oxygen penetration into the surface sediment of gravity core GeoB 15403-4 was identified by distinct color changes at 6 cm. The oxygen penetration depth in a multicore from the same station (GeoB15403-6) was identified at 8cm sediment depth. Furthermore, when the core was opened and sampled, the sediment surface appeared undisturbed with no indications of compression or sediment loss as seen on core pictures. Therefore, we argue that the core top represents the year 2011 when the core was retrieved. To support this, we correlated Ca/Ti data from the XRF core scanning to the adjacent sediment core DP30, which contains modern sediments at the core top as proven by [210]Pb dating (Goudeau et al., 2014) (RRC 3, Figure 1). Although accumulation rates at the two sites are different, the close correspondence of the Ca/Ti records (further validated by the available AMS [14]C ages) suggests undisturbed records and confirms the reliability of our age model. The presence of an undisturbed sediment surface in GeoB15403-4 is further supported by the continuation of the Ca/Ti curve at site GeoB15403-4 since it was sampled few years after core DP30. This information will be added to the manuscript appropriately.

[Figure]

**RRC3 Figure 1**: **Comparison of Ca/Ti records of adjacent cores DP30 (blue curve; Goudeau et al. 2014) and GeoB15403-4 (red curve). The close correspondence and available [210]Pb data for DP30 suggest the presence of a largely undisturbed sediment surface in both records.**

For further refinement of our age model of multicore GeoB15406-4 we now use the stable oxygen isotope record of the benthic foraminifer *Uvigerina mediterranea* for correlation with the [210]Pb dated multicore NU04 (Grauel et al., 2013). The isotope records show a close correspondence and the revised age model exhibits deviations in the range of -19 to +16 years (standard deviation of 12.2 years). We will use this new age model, which is still in accordance with the AMS [14]C age for the base of our multicore and does not change the general interpretation of our proxy data.

*3) The second problem is associated to the occurrence of a tephra layer related to Lipari Eruption. The authors reported as reference for this tephra, a manuscript submitted as Menke et al.. I do not think that the authors can use this tephra as tiepoint, reporting as reference a manuscript under review. In addition, the authors refer this tephra layer to Lipari eruption, but this eruption is related to Monte Pilato eruption phases (see Forni et al., 2013, Geological Society of London). This Monte Pilato volcanic phase spans from 729 to 1220 AD (see Forni et al., 2013, Geological Society of London). Why did the authors decide to associate this tephra layer to an age of 1174 yr BP (776 yr AD) that corresponds (or it is very close) to the age associated to the beginning of this eruption phase? Maybe it is wright, but if this reconstruction is based on data reported in the manuscript Menke et al. under review, I think that the authors have to improve the present version of the manuscript. Alternatively, they have to exclude this tephra from the present age model.*

It was unfavorable that the cited manuscript concerning the age model of GeoB15403-4 based on tephrostratigraphy was not available at the time of the review process. This manuscript is now accepted for publication in the March Issue of Quaternary Research (Volume 89, Issue 2). For a detailed description and critical discussion of the age dating of the ash layer see Menke et al. (2018) (https://doi.org/10.1017/qua.2018.2). Several

arguments favor the AD 776 Monte Pilato eruption rather than the AD 1220 Lami eruption. The Monte Pilato eruption is characterized by slightly higher $SiO_2$ and lower $K_2O$ contents, which is in accordance with the geochemical analyses of the detected cryptotephra. Based on associated pyroclastic layers, the size of the Monte Pilato eruption was much larger when compared to the Lami eruption. Accordingly, the Monte Pilato eruption had at least a subplinian character, which provided the necessary size to inject sufficient volcanic matter high into the atmosphere to be transported over the required distance to the Gulf of Taranto.

*4) The authors reported from line 5 to line 10 (pag 5) information concerning the undisturbed sediment in the uppermost part of the records. Without radionuclides and proxy of porosity, it is not possible to make this assumption. There no evidence to support this assumption. In table 1, it is necessary to specify if the authors run AMS14C on mix of planktonic foraminifera or on single species. In addition, it is important to indicate the thickness of the sample used for each AMS14C dating. This information is important to analyse the propagation of errors. In your age-depth profile (Fig. 2) it is important to show the propagation of errors. Because of the authors have no radionuclides data, the propagation of errors cannot be extended to top core. Please it is necessary to show the algorithm used to create the age model.*

Concerning the age model of the upper part of the core see our response to comment 2). As stated in the manuscript we used samples of mixed surface-dwelling planktonic foraminifera to generate the AMS $^{14}$C dates. These comprised shells of *Globigerinoides ruber* (variety pink and white), *Globigerinoides conglobatus*, *Globigerina bulloides*, *Orbulina universa* and *Globigerinella siphonifera*. We avoided shells of deep-dwelling taxa such as *Globorotalia inflata*, *Globorotalia truncatulinoides, and Neogloboquadrina dutertrei*. A description of the species that were used will be added to the methods section. Figure 2 B and C show the thickness of the depth interval that was used for dating with a horizontal line, however since the depth interval for site 03 is so small, the line in Figure 2 B is hard to see. We suggest to add a column with values for the sample thickness to Table 1.

*5) Spectral analyses: I would like to suggest to use the "Intrinsic Mode Functions" (IMF) (Huang et al., 1998) to analyse the signal and to run wavelet analysis on selected IMF component. This is the correct approach when you analysis records for the last millennia. Only with this approach you can identify the stable frequency associated to your proxy and if it is continuous present along the whole study record. The single spectrum reported in figure 11 represents a mean value within the record, so that it is not representative of a possible forcing.*

We agree that wavelet analysis will add significant information on the continuity of the periodic variations in our data. Accordingly, we will perform wavelet analyses on the SIIBF and EBF data series. We will use the approach of Torrence and Compo (1998) applying a Morlet wavelet spectrum. Both, Morlet wavelet spectrum and Hilbert spectrum reveal similar energy-frequency distributions. In addition, we will perform time-series analyses using the REDFIT algorithm of Schulz and Mudelsee (2002) because it allows a proper error estimation of the global spectrum.

*6) Concerning the recently scientific literature focused on the shallow water environment, the authors did not consider in this manuscript several articles (Ferraro et al., 2012; Vallefuoco et al., 2012; Lirer et al., 2014; Margaritelli et al., 2016; Di Rita et al., 2018; Oldfield et al. 2003; Bonomo et al., 2016; Di Bella et al., 2014). These articles in my opinion offer some important information for the submitted manuscript. Ferraro et al. (2012) and Di Bella et al. (2014) for benthic foraminifera, Bonomo et al. (2016) and Di Rita et al. (2018) relation between NAO and runoff/alboreal pollen, Oldfield et al. (2003) with low resolution concerning benthic foraminifera, etc. . ... Are there specific reasons for this choice? Concerning the NAO index, I*

*think that the article Brunetti et al (2002) focused on the winter precipitation in Italy modulated by NAO, has to be take in account. In addition, the NAO forcing has been shown also in other fossil marine sedimentary archives by several authors (Chen et al., 2011; Nieto-Moreno et al., 2013; Goudeau et al., 2015; Jalali et al., 2015). Why the authors did not consider these references from Mediterranean area?*

We appreciate the recommended literature and will consider the relevant references in appropriate sections of the manuscript.

*7) In Figure 3A, the authors compared SIBF signals in the two records, but as documented in the figure for the last two centuries, the two curves have an antithetic pattern. The same framework, maybe less pronounced, is also shown for SIIBF signals. In my opinion, in the manuscript the explanation reported for this discrepancy in the last two centuries is not scientific supported. In my opinion, without radionuclides dating this problem cannot be solved. Again, I would like to suggest to the authors to plot the distribution patterns of benthic foraminifera per gr of sediment to understand or to try to interpret correctly this discrepancy. In my opinion, the differences in benthic assemblage reported for both study sites in figure 6, is not so evident. In addition, without a detailed high-resolution morphobatimetry of the study area it is not possible to propose this type of interpretation.*

After applying the revised age models to our records the different trends in the abundance of SIIBF are still evident and can be attributed to spatial differences in organic matter flux rates. According to our experience, the benthic foraminiferal number BFN (Ind./g sediment) seems not appropriate to evaluate the microhabitat structure and thus ecological significance of the fauna. The BFN or BF accumulation rate has been interpreted as a general function of organic matter fluxes to the sea floor in the Pacific Ocean (Herguera and Berger, 1991). However the applicability of this approach in other oceans has been challenged since then (e.g., Schmiedl et al. 1997) because this parameter is strongly influenced by variable accumulation rates and other factors. Application of the BFN (to the bulk fauna and single taxa) appears useful when the counting sums are low in samples containing only few individuals and relative numbers would lead to misinterpretations. In our samples we have aimed at counting 300 individuals or more per sample, allowing a proper ecological evaluation as reflected by the relative proportion of foraminifera with different microhabitat preferences. Specifically, higher relative proportions of SIIBF clearly indicate increased organic matter fluxes and associated food availability at the sea-floor and vice versa.

*9) The authors have to focus as follows: 1) on chronology of the last two centuries and on the determination of propagation of errors, 2) on the interpretation of benthic foraminifera per gr of sediment and not in percentages, due to the target of the manuscript. This approach could help to interpret the benthic data vs the target of this manuscript. 3) I think that the authors have to take in account also the several dams build along the rivers of the Adriatic Sea. These constructions changed significantly the sediment outflow in Adriatic Sea. 4) I would like to suggest to see also the contribution of Ofanto river. 5) It is necessary to improve the spectral and wavelet analysis carried out on the proxies. 6) If is necessary to filter each frequency and compared these with internal or external forcing. 7) Due to the target of the manuscript it is necessary to compare the study records (biotic or abiotic proxies) with proxy of river discharge.*

We will revise our manuscript including most of the suggestions by the reviewer as outlined above. The chronology has been further tested and confirmed for GeoB15403-4 and will be refined for core GeoB15406-4. We will still base the ecological evaluation of the benthic foraminifera on the relative proportion of different microhabitat groups because the high counting sums guarantees a proper representation while the BFN can be biased by variable accumulation rates and other factors (such as patchiness). We will consider the other

mentioned points appropriately. Specifically we will add wavelet analyses and elaborate a more on the riverine contributions of the different areas.

Cited References:

Goudeau, M. L. S., Grauel, A. L., Tessarolo, C., Leider, A., Chen, L., Bernasconi, S. M., Versteegh, G. J. M., Zonneveld, K. A. F., Boer, W., Alonso-Hernandez, C. M., and De Lange, G. J., 2014, The Glacial-Interglacial transition and Holocene environmental changes in sediments from the Gulf of Taranto, central Mediterranean: Marine Geology, v. 348, p. 88-102.

Grauel, A. L., Leider, A., Goudeau, M. L. S., Muller, I. A., Bernasconi, S. M., Hinrichs, K. U., de Lange, G. J., Zonneveld, K. A. F., and Versteegh, G. J. M., 2013, What do SST proxies really tell us? A high-resolution multiproxy (U-37(K '), TEX86H and foraminifera delta O-18) study in the Gulf of Taranto, central Mediterranean Sea: Quaternary Science Reviews, v. 73, p. 115-131.

Herguera, J. C., and Berger, W., 1991, Paleoproductivity from benthic foraminifera abundance: Glacial to postglacial change in the west-equatorial Pacific: Geology, v. 19, no. 12, p. 1173-1176.

Schmiedl, G., and Mackensen, A., 1997, Late Quaternary paleoproductivity and deep water circulation in the eastern South Atlantic Ocean: Evidence from benthic foraminifera: Palaeogeography, Palaeoclimatology, Palaeoecology, v. 130, no. 1, p. 43-80.

Schulz, M., and Mudelsee, M., 2002, REDFIT: estimating red-noise spectra directly from unevenly spaced paleoclimatic time series: Computers & Geosciences, v. 28, no. 3, p. 421-426.

Torrence, C., and Compo, G. P., 1998, A practical guide to wavelet analysis: Bulletin of the American Meteorological society, v. 79, no. 1, p. 61-78.